# Unsupervised Similarity Learning for Spectral Clustering

## Abstract

Spectral clustering has been popularized due to its ability to identify non-convex boundaries between individual clusters. However, it requires defining a similarity metric to construct the Laplacian matrix. Instead of predefining this metric upfront, we propose to learn it by finding the optimal parameters of a kernel function. This learning approach parameterizes the data topology by optimizing a similarity function that assigns high similarity values to a pair of data that share discriminative features and vice versa. While some existing approaches also learn the similarity values, they rely on hyperparameters to do so. However, these hyperparameters cannot be validated in an unsupervised setting. As a result, suboptimal hyperparameter values can lead to detrimental performance. To circumvent this drawback, we propose a method that eliminates the need for hyperparameters by learning the optimal parameter for a similarity metric used in spectral clustering. This enables unsupervised learning of the similarity metric while performing spectral clustering. The method's capability is verified on several benchmark datasets with a large scale of non-convexity. Our method outperforms SOTA approaches on accuracy and normalized mutual information measures up to 10% when applied to popular image and text datasets.

## 1 Introduction

Spectral clustering (SC) (von Luxburg, 2007) is a very effective method to cluster data with non-convex separation boundaries. In this approach, the individual data points can be considered as nodes of a fully connected graph where the edge weights are the similarity values. The underlying assumption for SC is that for every pair of data in the training set, the similar ones should have a high similarity score and vice versa. Thus, the objective for SC becomes minimizing the sum of edge weight between clusters and maximizing the sum within clusters (see fig. 1).

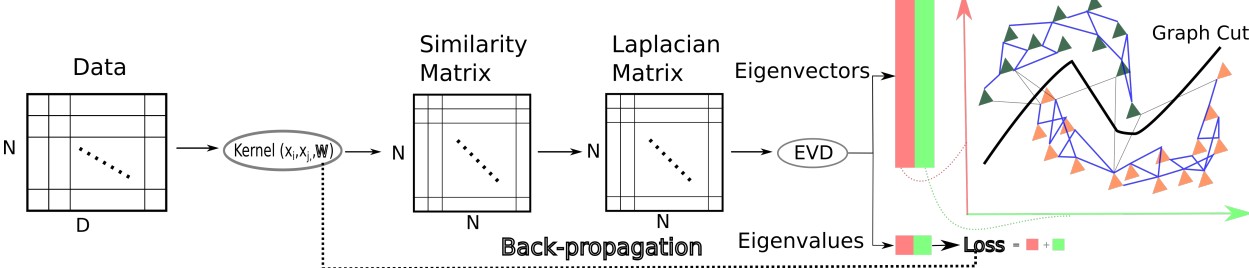

Figure 1: The parameters of the kernel function are learned through gradient descent, using the sum of the first $K$ eigenvalues as the loss function.

The performance of SC is primarily attributed to the manually chosen parameters of the *similarity metric*. However, an incorrectly chosen similarity metric can be detrimental to the clustering performance as it might not emphasize enough the similar pairs relative to the dissimilar ones (see fig. 2c). Hence, instead of hand-picking the parameters of the similarity metric, one can try to learn instead (see fig. 1).

Methods like Kang et al. (2017) aim to learn the similarity metric by defining an objective function. However, it features one or more hyperparameters in the objective function. Unlike supervised learning, validating

hyperparameter values through an annotated validation set is not possible in an unsupervised learning setup. Given the need for an extensive exploratory and exploitative search in supervised hyperparameter optimization, it is unlikely that optimal parameters will be selected due to the suboptimal range of values being considerably larger than the optimal range in real-world scenarios.

If we apply the same principle to an unsupervised setting, randomly selecting hyperparameters would likely result in suboptimal values. This, in turn, would result in incorrect similarity scores affecting the downstream clustering performance (Fan et al.). To mitigate this drawback, we propose learning a parameterized similarity metric without introducing any hyperparameter. To do so, *we assume the similarity metric to be a radial basis function (RBF) kernel and try to find its optimal bandwidth.* Since the RBF kernel projects the data into the infinite-dimensional space (Shashua, 2009), its bandwidth enables modulation of the most important dimensions. Moreover, the parameter search space reduces from the number of every possible data pair (when no assumption is made) to the number of parameters in the kernel function.

In the reduced parameter space, the search for the optimal parameter of the similarity metric involves the sum of the first $K$ eigenvalues from the Laplacian matrix (refer to fig. 1). In this study, the initial number of clusters (i.e., K) is predetermined; however, the unsupervised estimation of the cluster count is possible by identifying density peaks within the data topology, as demonstrated in Rodriguez & Laio (2014). We find the optimal parameter of the kernel using gradient descent is possible by back-propagation through the eigenvalue decomposition (EVD) (see fig. 1).

Building on these insights, the method we introduced offers an unsupervised approach to learning similarities that determine the optimal weights for the edges of a fully connected graph, on which SC is then applied. We assessed the clustering performance of our method on various image and text datasets, demonstrating an improvement over alternative approaches.

The contributions of this work are the following.
1. We introduce a novel unsupervised approach to similarity learning aimed at optimizing spectral clustering.
2. This technique has been tested and proven to provide improved clustering performance on standard benchmark datasets for images and texts.

The proposed method learns the parameter governing the similarity metric and mitigates the need for auxiliary hyperparameters. By learning instead of handpicking the parameters of the similarity metric one can guarantee a better data-driven clustering performance.

## 2 Method

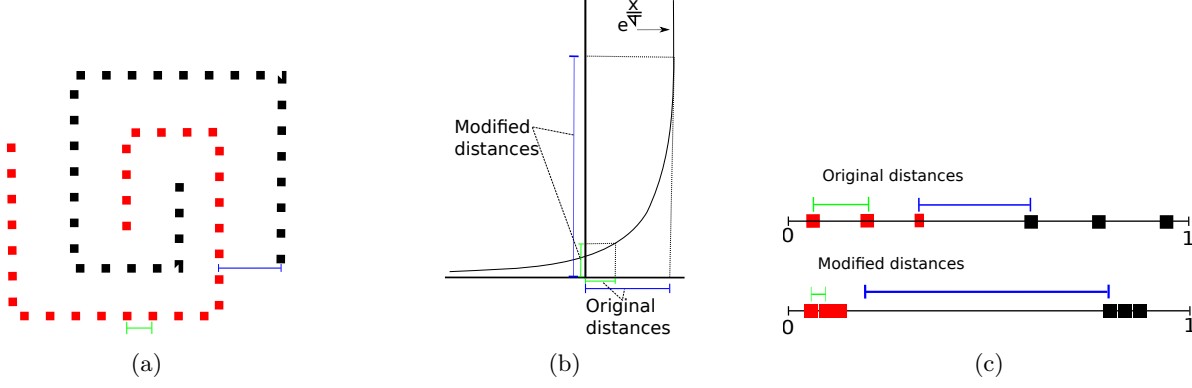

(a)            (b)            (c)

Figure 2: From the given data in fig. 2a, the gap between each cluster is amplified. To amplify the gaps between each cluster, every distance is modulated using the exponential function of the RBF kernel fig. 2b. As an effect of this, bigger distances are made even bigger, and smaller distances are made even smaller after normalization, as shown in fig. 2c. Upon normalization to a unit vector, the magnitude of smaller distances is reduced, whereas that of larger ones is increased fig. 2c.

The proposed method works directly on the raw data (e.g., data with the topology as in fig. 2a) and tries to further emphasize any existing separation within the data points (see fig. 1). Accordingly, the optimal similarity function would lower the smaller pairwise distances so that all the data points would eventually collapse to a single point (see fig. 2c). At the same time, the more pronounced distances are amplified (see fig. 2c). To do so, we choose to exponentiate these pairwise distances in a controlled manner (see eq. (1) and fig. 2b), where the smaller the distance between two data points, the higher their similarity and vice versa. As a result, the bigger distances are exponentiated towards higher values much more than the smaller ones (see fig. 2b). Since the new transformed distances are normalized within a unit vector, increasing the bigger magnitudes would render the smaller ones towards even smaller values and vice versa (see fig. 2c).

The RBF kernel[1][2] enables a controlled exponential modulation of the distances via its bandwidth $\sigma$ (see eq. (1)).

$$K(x,y) = e^{\frac{-d(x,y)}{\sigma^2}} = \sum_{n=0}^{\infty} \frac{-d^n(x,y)}{(\sigma^2)^n n!} \tag{1}$$

The popularity of RBF is attributed to its ability to map data to infinity-dimensional space (Shashua, 2009). By properly tuning the RBF bandwidth $\sigma$, one can smoothly truncate the infinite-dimensional expansion (see eq. (1)) and select the dimensions to be incorporated in the comparison. In other words, $\sigma$ adjusts the amount of modulation for the similarity metric. Using the RBF kernel as a generic similarity formulation, the task reduces to optimization of $\sigma$ that yields the most representative distance from a general formulation. To amplify the distance $d(x,y)$ as presented in eq. (1), the parameter $\sigma$ must be set to a value less than one (i.e., $\sigma \leq 1$) where $\sigma = 0$ is not allowed. *Traditionally, in a spectral clustering setting, $\sigma$ is a hyper-parameter which is handpicked. Instead, we propose learning $\sigma$ directly from the given data.*

### 2.1 Learning operation

In SC, the objective remains identical to other clustering techniques; similar data are projected together while the dissimilar ones are far apart (von Luxburg, 2007). This objective is equivalent to the GC (see fig. 1) when data are considered nodes of the connected graph while the edges are the pair similarity values. Thus, to construct an objective function that is amenable to the gradient-based method, one has to consider the data projections (i.e., $\boldsymbol{f}$) and the similarity values ($A_{i,j} \in \mathbb{R}, \forall i,j \in \{1,..,N\}$) simultaneously. Utilizing the general formulation for GC (von Luxburg, 2007) (see eq. (2)) as the objective function, it is possible to get an empirical assessment of clustering performance, where $\boldsymbol{L}$ in eq. (2) is the Laplacian matrix and $\boldsymbol{f}$ are the eigenvalues.

$$\text{GC}[\boldsymbol{A}, \boldsymbol{f}] = \sum_{i,j,j<i} A_{i,j}(f_i - f_j)^2 = \boldsymbol{f}^T \boldsymbol{L} \boldsymbol{f}. \tag{2}$$

The formulation in eq. (2) reduces the task to learning the similarity matrix (i.e., $\boldsymbol{A}^{[N,N]}$) and the representation of the data (i.e., $\boldsymbol{f}$). As a result, the computation of GC on the graph reduces to matrix multiplication. Since the utilized similarity is parameterized solely by the RBF bandwidth $\sigma$, we optimize the computed similarity matrix by tuning $\sigma$ (see eq. (3)).

$$\arg\min_{\sigma,\boldsymbol{f}} \text{GC}[\boldsymbol{A}(\sigma), \boldsymbol{f}(\boldsymbol{A})] \rightarrow \frac{\partial^2 \text{GC}[\boldsymbol{A}(\sigma), \boldsymbol{f}]}{\partial\sigma\partial\boldsymbol{f}} = 0. \tag{3}$$

Instead of finding both $\sigma$ and $\boldsymbol{f}$ simultaneously, an alternation between each parameter is adopted (see algorithm 1). We utilize a Lagrangian multiplier to restrict the domain of the data representation $\boldsymbol{f}$ to a unit magnitude $\boldsymbol{f}\boldsymbol{f}^T = 1$ (see eq. (4) from Strang (2006)).

---

[1]To make the metric entirely data-driven, the factor of two that should be in the denominator of the exponent in eq. (1) has been excluded.

[2]$d(x,y)$ in eq. (1) represent the Euclidean distance.

$$\frac{\partial\{\mathrm{GC}[\boldsymbol{A}(\sigma),\boldsymbol{f}]+\lambda(\boldsymbol{f}^T\boldsymbol{f}-1)\}}{\partial\boldsymbol{f}}=0\rightarrow\boldsymbol{L}\boldsymbol{f}=\lambda\boldsymbol{f} \tag{4}$$

Using eq. (4) from Strang (2006), one can derive $\boldsymbol{f}$ and $\lambda$ as the eigenvectors and eigenvalue of the Laplacian matrix $\boldsymbol{L}$. Since the RBF kernel is symmetric (see eq. (1)), the resulting Laplacian matrix L is also symmetric upon which EVD produces real and non-negative eigenvalues ($\lambda_i \geq 0, \forall i \in 1,..,N$), and orthonormal eigenvectors ($\boldsymbol{f}^T\boldsymbol{f}=1$) (Strang, 2006). Utilizing the computed data representation ($\boldsymbol{f}$), it is possible to rewrite GC as in eq. (5) from Strang (2006).

$$\mathrm{GC}[\boldsymbol{A}(\sigma),\boldsymbol{f}(A)]=\boldsymbol{f}^T\boldsymbol{L}(\sigma)\boldsymbol{f}=\boldsymbol{f}^T\lambda(\sigma)\boldsymbol{f}=\lambda(\sigma)\underbrace{\boldsymbol{f}^T\boldsymbol{f}}_{\boldsymbol{f}^T\boldsymbol{f}=1}=\lambda(\sigma) \tag{5}$$

In the case of multiple different orthonormal eigenvector embeddings, the GC reduces to the sum of the corresponding eigenvalues $\lambda$ as in eq. (6) (as indicated by Ky Fan's Theorem (So et al., 2010)):

$$\mathrm{GC}[\boldsymbol{A}(\sigma),f_{i,..,K}(A)]=\mathrm{GC}(\sigma)|_{f_{i,..,K}}=\sum_{i=1}^{K}\lambda_i(\sigma) \tag{6}$$

Whereby, we incrementally fine-tuned $\sigma$ using discrete steps of magnitude $\Delta\sigma$, which were determined through a linear approximation near $\sigma$ for $\sum_{i=1}^{K}\lambda_i(\sigma)$ (see eq. (7)). Optimal adjustment of $\sigma$ is achieved when we choose $\Delta\sigma$ to be in the opposite direction to $\frac{\partial\sum_{i=1}^{K}\lambda_i(\sigma)}{\partial\sigma}$ (see eq. (8)).

$$\sum_{i=1}^{K}\lambda_i(\sigma+\Delta\sigma)\approx\sum_{i=1}^{K}\lambda_i(\sigma)+\Delta\sigma\frac{\partial\sum_{i=1}^{K}\lambda_i(\sigma)}{\partial\sigma} \tag{7}$$

$$\Delta\sigma=-\underbrace{\eta}_{\text{Learning rate}}\frac{\partial\sum_{i=1}^{K}\lambda_i(\sigma)}{\partial\sigma}\Rightarrow\underbrace{\sigma_i\leftarrow\sigma_i-\eta\frac{\partial\sum_{i=1}^{K}\lambda_i(\sigma)}{\partial\sigma}}_{\text{Gradient descend on }\sigma} \tag{8}$$

The new algorithm, built on top of the SC setup (von Luxburg, 2007), adopts a two-stage operation within each iteration, starting by vanilla SC setup followed by the $\sigma$ update (see algorithm 1). In the initial phase, the parameter $\sigma$ is held constant, allowing for the computation of the first $K$ eigenvectors $\boldsymbol{f}$, which serve as the embeddings for the dataset. Subsequently, the second phase maintains these computed eigenvectors $\boldsymbol{f}$ fixed while it refines the value of $\sigma$.

To introduce the scale invariance in the Euclidean distance matrix ($\boldsymbol{E}^{[N,N]}$ in algorithm 1), its values are normalized so that its minimum value becomes zero and its maximum value becomes one. As the denominator involves the square of $\sigma$, both positive and negative values of $\sigma$ of the same magnitude have the same modulation effect in the RBF kernel eq. (1). Hence, we consider only positive values for $\sigma$, as its negative values do not introduce any extra impact. To mitigate the impact of the initialization of the learning rate, its initial value is set to one (i.e., $lr=1$). Consequently, the initial update on $\sigma$ is driven solely by its gradient. We decimate the learning rate if $\sigma$ assumes a value that is not in $\Sigma=(0,1]$.

The parameter $\sigma$ is inherently data-dependent, adapting its value to the specifics of the dataset under analysis. The gradient of $\sigma$ acts as a guiding mechanism, tending toward zero as $\sigma$ approaches the optimal point. Notice that the gradient of the loss (i.e., $\frac{\partial\mathrm{loss}(\sigma_i)}{\partial\sigma_i}\geq 0, \forall\sigma_i\in\mathbb{R}_{\geq 0}$ in algorithm 1) is always non-negative within the domain of positive real numbers (see theorem 1 in Appendix) hence $\sigma$ is always non-increasing in $\Sigma=(0,1]$. Therefore, the goal of the optimization process is to minimize $\sigma$ until its gradient approaches a negligibly small magnitude.

---

**Algorithm 1** Similarity Learning

---

1: **procedure** INPUT:(Data $\boldsymbol{X}^{[N,D]}$, Number of clusters $K$)
2:    Compute the Euclidean distance matrix $\boldsymbol{E}^{[N,N]}$.
3:    Scale the Euclidean distance matrix: $\boldsymbol{E}^{[N,N]} = \frac{\boldsymbol{E}^{[N,N]}}{\max \boldsymbol{E}^{[N,N]}}$.
4:    $\sigma_0 = 1$.                              ▷ No modulation effect for RBF kernel as a start.
5:    $lr = 1$.                                      ▷ Initially, only the gradient updates $\sigma$.
6:    **for** each $i = 1, 2, \ldots n_{iter}$ **do**
7:        ***Vanilla SC Setup***
8:        Affinity Matrix: $\boldsymbol{A}^{[N,N]} \leftarrow e^{-\frac{\boldsymbol{E}^{2[N,N]}}{\sigma_i^2}}$.
9:        Degree Matrix: $\boldsymbol{D}^{[j,j]} \leftarrow \sum_{i=0}^{N} \boldsymbol{A}^{[i,j]}, \forall j \in \{1,..,N\}$.
10:       Laplacian Matrix: $\boldsymbol{L}^{[N,N]} \leftarrow \boldsymbol{D}^{[N,N]} - \boldsymbol{A}^{[N,N]}$.
11:       First K eigenvalues: $\lambda_{1,..,K} \leftarrow \text{EVD}(\boldsymbol{L})$
12:       ***Our proposal for $\sigma$ learning***
13:       Compute: $\text{loss}(\sigma_i) \leftarrow \sum_{i=1}^{K} \lambda_i(\sigma_i)$
14:       Back-propagation: $\frac{\partial \text{loss}(\sigma_i)}{\partial \sigma_i}$
15:       Gradient descent: $\sigma_i \leftarrow \sigma_i - lr \frac{\partial \text{loss}(\sigma_i)}{\partial \sigma_i}$
16:       **if** $\sigma_i \notin \Sigma$ **then**                    ▷ Negative $\sigma$ are excluded.
17:           $\sigma_i \leftarrow \sigma_i + lr \times \frac{\partial \text{loss}(\sigma_i)}{\partial \sigma}$              ▷ Correct the $\sigma_i$.
18:           $lr \leftarrow lr/10$                              ▷ Decimate the $lr$.
19:    **return** $\sigma$                                ▷ Return the learned parameter $\sigma$.

---

Although reducing the magnitude of $\sigma$ does lead to a lower graph cut value in spectral clustering, setting $\sigma$ too close to zero is not advisable. This setting results in the edge weights of the graph converging to zero. Consequently, each data point becomes isolated into its own cluster, effectively obliterating the underlying data structure and preventing the detection of inherent clusters.

This proposed method works solely on the Laplacian matrix setup. Other versions of normalized Laplacian were tried, i.e., symmetric normalized Laplacian $\left(\boldsymbol{L}_{\text{normal}}^{\text{symmetric}} = \boldsymbol{D}^{-\frac{1}{2}} \boldsymbol{L} \boldsymbol{D}^{-1\frac{1}{2}}\right)$, and normalized Laplacian row-wise stochastic version $\left(\boldsymbol{L}_{\text{normal}}^{\text{stochastic}} = \boldsymbol{D}^{-1} \boldsymbol{L}\right)$ but the gradient computation was not updating $\sigma$ in the anticipated direction. Furthermore, different kernels did not perform on the setup; therefore, *the RBF kernel is the only one that enables the update of the parameters within the Laplacian setup.*

In our method, we strategically reduce the RBF bandwidth towards zero, enabling SC to adhere to the maximum margin principle (Hofmeyr, 2020). This strategy, however, encounters a challenge when isolated data points, distant from the main data body, form their "singleton" clusters (Hofmeyr, 2020).

Since the eigenvectors enable a bi-partition on the dataset with a GC cost indicated by its corresponding eigenvalues (von Luxburg, 2007), the singletons, which are notably detached from the primary dataset, can be effectively separated using a GC that harnesses the eigenvector corresponding to the smallest eigenvalue (Hofmeyr, 2020). This separation mechanism is scalable to multiple outliers, allowing their separation through the initial set of eigenvectors equivalent in number to the outliers.

Given that, these singleton occurrences can be easily identified by sequentially applying GC on the foremost eigenvector, the count of these vectors directly represents the number of singletons. Consequently, once we have identified the singletons, we utilize the subsequent eigenvectors to partition the remainder of the dataset into non-singleton clusters. Furthermore, we experimented on different datasets containing outliers and showcased that method's ability to perform well.

**Complexity analysis:** The Euclidean distance matrix is computed once outside the loop at $O(N^2)$ complexity. Since we are looking solely for the first $K$ eigenvectors, the EVD complexity is $O(K \times N^2)$ (Golub & Van Loan). Similarly, the complexity of gradient computation using the back-propagation is $O(K \times N^2)$ since the output of the computational graph has $K$ eigenvalues (Baydin et al.). Over $n_{iter}$ iterations, the total complexity would be $O(n_{iter} \times K \times N^2)$.

## 3 Related work

The similarity learning domain contains various methodologies established on different fundamental assumptions (Qiao et al., 2018).

**Empirically heuristic:** The heuristic methods assigns a unique $\sigma$ for each data point by considering local statistics within a selected neighborhood in the raw (Zelnik-manor & Perona, 2004) or embedded space (Ng et al., 2001). Yet, scaling this approach is difficult because it requires computing as many $\sigma$ values as the dataset size, and the choice of neighborhood critically affects performance.

**Graph learning:** Rather than relying on heuristics, some studies adopt a data-driven approach to individually learn a scale parameter, (i.e., $\{\sigma_k\}_{k=1}^d$, where $d$ is the number of dimensions), for each dimension of the dataset (Karasuyama & Mamitsuka, 2013). The learning process involves minimizing the cumulative distance between each data point $x_i$ and the weighted mean of its neighbors as in eq. (9):

$$\min_{\{\sigma_k\}_{k=1}^d} \sum_{i=1}^n \left\| x_i - \frac{1}{d_j} \sum_{j=1}^n E_{ij} W_{ij} x_j \right\|, \text{ s.t: } d_j \to \text{node degree} \tag{9}$$

In eq. (9), $W_{ij}$ represents the affinity between data points, defined as $W_{ij} = \exp\left(-\frac{1}{2}\sum_{k=1}^d \frac{d^3(x_{ik}, x_{jk})}{\sigma_k^2}\right)$, and $E_{ij}$ denotes the elements of the connectivity matrix $\boldsymbol{E}$ (filled with binary values to indicate the presence (i.e., $E_{ij} = 1$) or absence (i.e., $E_{ij} = 0$) of an edge between each data pair), which are determined separately using a heuristic. The values of $\{\sigma_k\}_{k=1}^d$ are optimized through a gradient descent procedure, iteratively refining them to improve the representation of the manifold's local geometry.

Certain techniques induce locality constraints to evaluate similarity values (i.e., $\boldsymbol{W}$) (Roweis & Saul, 2000). This is achieved by minimizing the cumulative difference of individual data relative to neighborhood constraints as in eq. (10).

$$\min_{\boldsymbol{W}} \sum_{i=1}^n \left\| x_i - \sum_{j=1}^n E_{ij} W_{ij} x_j \right\|, \text{ s.t: } \sum_{j=1}^n E_{ij} W_{ij} x_j = 1 \tag{10}$$

Further enhancements to this method include the imposition of an additional constraint on the similarity values, (i.e., $W_{ij} \geq 0$) (Wang & Zhang, 2008).

To overcome the limitations associated with heuristic-based methods for establishing the connectivity matrix (i.e., $E$), alternative methods have been developed that optimize the weight matrix $\boldsymbol{W}$ directly, bypassing the need for defining $\boldsymbol{E}$ (Zhang et al., 2014). The objective (see eq. (11)) ensures a smooth transition between nearby points, balanced clustering, and uniform edge weight distribution[4].

$$\min_{W \geq 0} \underbrace{\sum_{i,j=1}^n \|x_i - x_j\| W_{ij}}_{\text{Smooth transition}} + \underbrace{\lambda_1 \sum_{j=1}^n (d_j - 1)^2}_{\text{Balanced clustering}} + \underbrace{\lambda_2 \sum_{j=1}^n W_{ij}^2}_{\text{Uniform edge weight distribution}} \tag{11}$$

In comparison to methods that focus on local attributes (Zhang et al., 2014), a global model (Daitch et al., 2009) tries to fit a graph to a vector of data using the following quadratic objective eq. (12).

$$\min_{\boldsymbol{W}} \sum_{i=1}^n \left\| d_j x_i - \sum_{j \neq i}^n W_{ij} x_j \right\|^2 \text{ s.t: } d_j \geq 1 \text{ and } \sum_i (max(0, 1 - d_j))^2 \leq \eta n \tag{12}$$

---

[4]$\lambda_1, \lambda_2, \eta$ in eqs. (11) to (14) are regularizer hyperparameter coefficient.

An alternate methodology (Elhamifar & Vidal, 2012), tries to fit a graph to a specified data set by enforcing sparsity on the learned similarity values via $L_1$ regularization term (see eq. (13)).

$$\min_{W} \left( \|\boldsymbol{X} - \boldsymbol{X}\boldsymbol{W}\|_F^2 + \lambda_1 \|\boldsymbol{W}\|_1 \right) \tag{13}$$

In contrast, some researchers have used a low-rank constraint by replacing the sparsity term (i.e., the $L_1$ norm, $\|\boldsymbol{W}\|_1$) with a low-rank term (i.e., the nuclear norm, $\|\boldsymbol{W}\|_*$) (Liu et al., 2010). While other approaches have even combined the low-rank constraint with sparsity (Zhuang et al., 2011), local constraint (Lu et al., 2013), Markov random walk (Liu et al., 2014), B-matching (Li & Fu, 2013; 2014).

Last but not least, instead of fitting a graph directly on the raw data, one can try to alternate between learning a linear transformation matrix (i.e., $\boldsymbol{P}$ in eq. (14)) of the data such that their pairwise comparison is more robust to noise and learning the similarity values (i.e., $\boldsymbol{W}$ eq. (14)) (Zhang et al., 2010) using the locality-preserving principle.

$$\min_{P,\boldsymbol{W}} \sum_{i,j=1}^{n} \|\boldsymbol{P}x_i - \boldsymbol{P}x_j\|^2 W_{ij} + \lambda_1 \sum_{i,j=1}^{n} W_{ij} ln(W_{ij}), \text{ s.t: } \sum_{i=1}^{n} \|\boldsymbol{P}x_i\|^2 = 1, \text{ and } \sum_{j=1}^{n} W_{ij} = 1, W_{ij} \geq 0. \tag{14}$$

Since $\|\boldsymbol{P}x_i - \boldsymbol{P}x_j\|^2 = \|x_i - x_j\| \boldsymbol{P}^T\boldsymbol{P} \|x_i - x_j\|$ one can consider that this method is unsupervised Mahalanobis distance learning.

Unlike existing methods, our approach is the first to explore learning a similarity metric for SC without adding any hyperparameters to either the similarity measure or the loss function.

## 4  Experiments

We ran a series of experiments on benchmark diagnostic datasets (see figs. 3 and 6 to 10 in Appendix) (Fränti & Sieranoja) as well as real-world datasets (see table 1) to showcase the method's capabilities at separating highly non-convex shapes (see fig. 3a and figs. 6a, 7a, 8a, 9a and 10a in Appendix) into linearly separable clusters (see fig. 3b and figs. 6b, 7b, 8b, 9b and 10b in Appendix). In all scenarios, the method tries to amplify the big pairwise distances within a confined space while reducing the small pairwise distances. Hence, the method finds a persisting gap within the topology of the dataset and tries to linearize it in the embedding space (see fig. 3b and figs. 6b, 7b, 8b, 9b and 10b in Appendix). One can notice from the optimization trajectory of the gradient for the RBF bandwidth $\sigma$ to converge towards zero (see fig. 3d and figs. 6d, 7d, 8d, 9d, 10d, 11d, 12d, 13d, 14d, 15d, 16b, 17b, 18b, 19b, 20b, 21b and 22b in Appendix).

To demonstrate the method's capability of managing isolated data points, we augmented each benchmark dataset (Fränti & Sieranoja) by adding two isolated points (see fig. 4a). During our experiments with these enhanced datasets, we modified the loss function to include additional eigenvalues equivalent to the number of isolated points (i.e., loss $= \sum_{i=1}^{K} \lambda_i$). For visualization, we omitted the first two eigenvectors since they only separate the singletons and are ineffective in separating the spiral patterns, focusing instead on the subsequent eigenvectors (see figs. 4 and 11 to 15 in Appendix) .

In the presented cases, the magnitude of $\sigma$ is progressively reduced until its gradient approaches zero. Furthermore, the cumulative magnitude of the first $K$ eigenvalues (i.e., loss $= \sum_{i=1}^{K} \lambda_i$) decreases throughout the training process (see figs. 3c and 5b and figs. 6c, 7c, 8c, 9c, 10c, 11c, 12c, 13c, 14c, 15c, 16a, 17a, 18a, 19a, 20a, 21a and 22a in Appendix). Notice that the smaller the magnitude of the first $K$ eigenvalues the more distinct the first $K$ clusters are in a dataset.

Furthermore, this linearization of the separation boundary reduces the complexity of the downstream tasks one can perform upon these embeddings. Clustering is one popular unsupervised task whose performance is solely the capability for learning the best representation of the data topology (von Luxburg, 2007). Hence, to

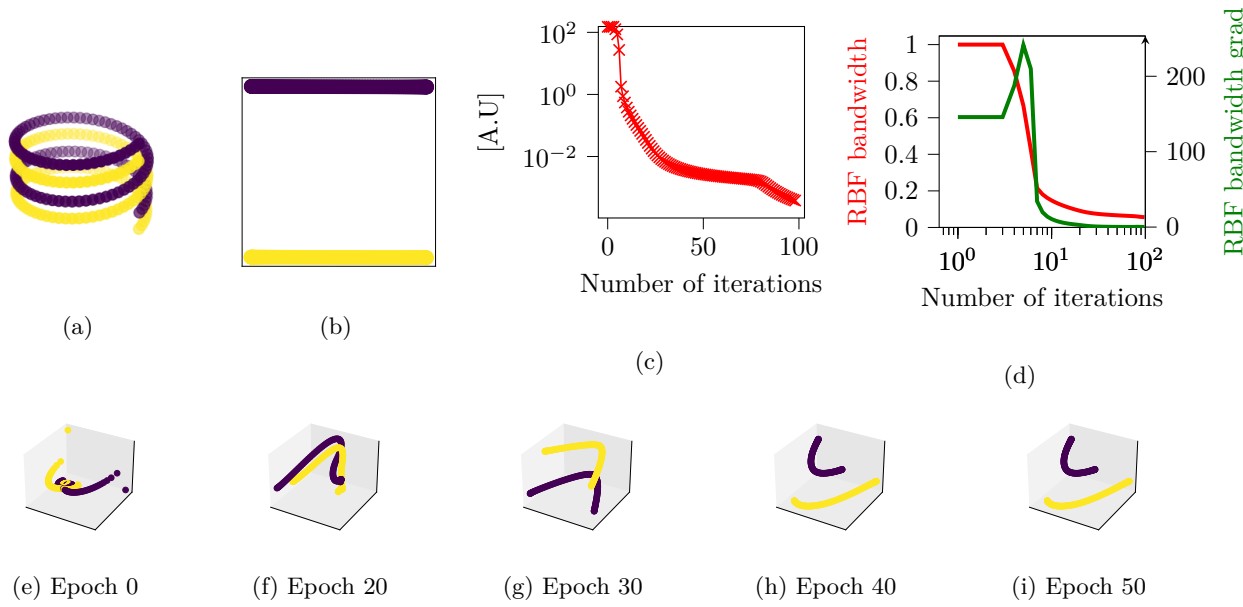

Figure 3: Benchmark dataset in fig. 3a. Our method learns a linear separation of these two spirals fig. 3b. In fig. 3b is the eigen-embedding of the benchmark data. In fig. 3c, is the loss function (loss $= \sum_{i=1}^{K} \lambda_i$) over during training. over iterations. The optimization of kernel bandwidth $\sigma$ using the gradient descent stabilizes as the $\sigma$ gradient diminishes fig. 3d. A snapshot of the optimized trajectory is in figs. 3e to 3i

Table 1: Description of the experimented datasets.

| Dataset | Size | Nr of Dimensions | Nr of Classes |
|---------|------|------------------|---------------|
| Yale    | 165  | 1024             | 15            |
| Jaffe   | 213  | 676              | 10            |
| ORL     | 400  | 1024             | 40            |
| COIL20  | 1440 | 1024             | 20            |
| BA      | 1404 | 320              | 36            |
| TR11    | 414  | 6429             | 9             |
| TR41    | 878  | 7454             | 10            |
| TR45    | 690  | 8261             | 10            |

compare the capability of our unsupervised method, we run K-means++ (Jain, 2010; Arthur & Vassilvitskii) on the eigen-embeddings where the clustering performance would be a proxy evaluator for the learned similarity matrix. We extend to realistic datasets where the models try to find linear separation between data clusters. In such a real data scenario, the proposed method tries to separate the dataset using persistent low-density (i.e., gap) regions throughout the dataset.

**Dataset description:** The datasets described in table 1 are widely utilized to assess clustering performance as no registration is needed upfront. The first three, JAFFE (Lyons et al., 1998), YALE (of Yale, 1997), ORL (Cambridge, 1994), contain human faces obtained at different illumination conditions, or different facial expressions, or with and without glasses. The last image dataset, COIL20 (Nene et al., 1996)) contains images of toys acquired at different orientations. While the rest of the data (BA, TR11, TR41, TR45) are text corpus (TR). These datasets are curated particularly for clustering algorithms that do not rely on deep learning techniques. Manual pre-registration of the images eliminates the need for translation invariance representation, which is typically achieved through the application of deep clustering techniques. Consequently, these datasets are suitable for testing traditional clustering methods that do not involve CNN-based feature extraction.

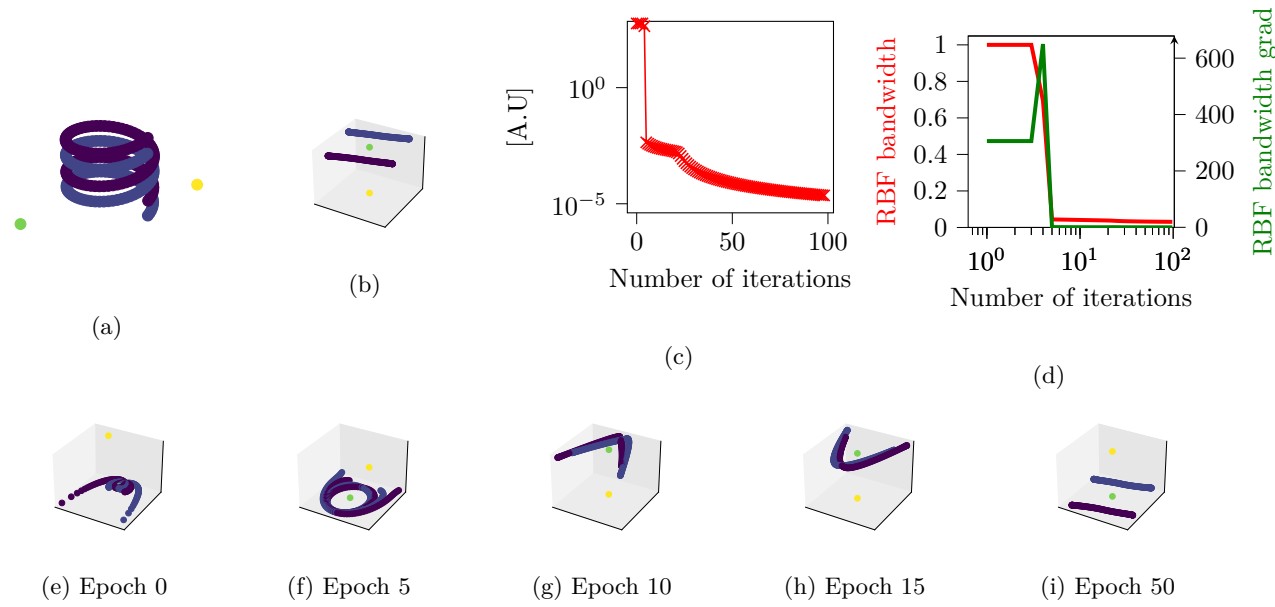

(a)

(b)

(c)

(d)

(e) Epoch 0    (f) Epoch 5    (g) Epoch 10    (h) Epoch 15    (i) Epoch 50

Figure 4: Our methodology successfully differentiates the two intertwined spirals despite the presence of two outliers, as depicted in fig. 4a, by learning a linear separation demonstrated in fig. 4b. The eigen-embedding of the benchmark data, shown in fig. 4c, reveals the intrinsic structure captured by our approach. The progression of the loss function (loss $= \sum_{i=1}^{K} \lambda_i$) over training iterations is depicted in fig. 4b, illustrating the convergence pattern. Our optimization process includes tuning the kernel bandwidth $\sigma$ via gradient descent, as shown in fig. 4c. Furthermore, a series of snapshots portraying the optimized separation trajectory can be seen in figs. 4e to 4i, where the first two eigenvectors (corresponding to the outliers) are omitted and the subsequent ones are visualized instead.

**Data preparation:** The image datasets are assumed to be correctly registered beforehand; therefore, no feature selection is needed upfront. To ensure the equivariance of the scale of the features, standardization ($\hat{x}_{\text{stand}} = \frac{x-\hat{\mu}}{\sqrt{\hat{\sigma}^2+\epsilon}}$) is performed before the computation of the Euclidean distance (see algorithm 1).

Once the similarity learning has been completed, we perform K-means++ (Jain, 2010; Arthur & Vassilvitskii) clustering upon the eigen-embeddings. To diminish the effect of the initialization, K-means + + has been run 50 different times until convergence.

Unlike our proposed method, *the competing ones have a hyperparameter(s) dependency that cannot be objectively tuned.* Hence, for a fairer comparison with our method, we report the performance of the competing method as the average across a range of hyperparameters. Moreover, we also report the maximal performance figures of the alternative methods (see table 2). Two different kernels of choice have been tested across 12 different hyperparameter combinations. RBF kernel eq. (1) across different bandwidth ($\{0.01, 0.05, 0.1, 1, 10, 50, 100\}$) along with a polynomial kernels $K(x, y) = (a + x^T y)^b$ with four different hyperparameter values (i.e., $a = \{0, 1\}$ and $b = \{2, 4\}$). The proposed method is entirely reproducible; therefore, we run just one experiment per dataset and report just one evaluation metric.

**Competing methods:** In table 2 we assess the effectiveness of the proposed method in a clustering scenario. Therefore we employ an SC setup in which the spectral embeddings derived from the acquired similarity matrix exhibit better performance compared to standard clustering methods. As competing methods, we utilize a range of clustering methods that incorporate kernel tricks with various hyperparameters, as well as kernel-free similarity learning methods.

Kernel K-means (KKM) (Dhillon et al., 2004) can perform the mean computation in the kernel reproducible space. This method uses a single kernel and demands arbitrary hyperparameter settings that must be set manually. Robust Kernel K-Means (RKKM) (Du et al., 2015) tries to attain maximum clustering

performance by combining multiple kernels on a K-Means setup. Multiple kernel k-means (MKKM) (Huang et al., 2012b) is similar to KKM while aggregating multiple kernels to attain a compounding effect in the final clustering performance.

Spectral Clustering (SC) (Ng et al., 2001) is the first method in the evaluation. Since our proposal is essentially built on top of SC, we can show that it is possible to optimize the RBF kernel, given the data. Therefore, vanilla SC serves as the baseline for the proposed method. Affinity aggregation spectral clustering (AASC) algorithm (Huang et al., 2012a) tries to consolidate multiple affinities matrix to refine the clustering result. Twin Learning for Similarity and Clustering (TLSC) (Kang et al., 2017) is another approach that does not require any kernel function but directly learns the similarity matrix and the clustering indexing. Similarity Learning via Kernel Preserving Embeddings (SLKE) (Kang et al., 2019) is similar to TLSC, which does not learn clustering indicator vectors but solely the similarity matrix. The SLKE algorithm has been experimented with in two distinct regularizations: Sparse SLKE (SLKE-S) and Low-ranked SLKE (SLKE-R). SC is performed on top of the learned similarity matrix via SLKE.

In table 3 compares our approach with two graph-based learning methods: Sparse Subspace Clustering (SSC) (Elhamifar & Vidal, 2012) and Low-Rank Representation (LRR) (Elhamifar & Vidal, 2012). Additionally, we consider six models that integrate graph construction with spectral embedding. These include Clustering with Adapting Neighbors (CAN) (Nie et al., 2014) and its variations: Projected Clustering with Adapting Neighbors (PCAN) (Yang et al., 2022) and Self-Weighted Clustering with Adapting Neighbors (SWCAN) (Nie et al., 2020). Additionally, we assess several other methods in our evaluation: LAPIN, which optimizes a bipartite graph for subspace clustering (Nie et al., 2023); DOGC, which learns the clustering discretization and graph learning simultaneously (Han et al., 2020); JGSED, a recent technique that integrates graph construction, spectral embedding, and cluster discretization (Qiao et al., 2018); and JSESR, an approach that combines spectral embedding with spectral rotation (Pang et al., 2020). Last but not least, self-tuning spectral clustering (SelfT) (Zelnik-manor & Perona, 2004) is also included in the comparison.

The performance of all these methods is conditional not only on the hyperparameters but also on the type of regularization. *The aim is to demonstrate the superiority of the proposed method in terms of clustering performance and ease of use.*

Recent advancements in deep clustering methods have demonstrated good performance on well-established image datasets. These methods comprise two stages: unsupervised representation learning, followed by the actual clustering step (Ren et al., 2022; Zhou et al., 2022). The bulk of the research in this area concentrates on representation learning. This phase involves organizing the data such that similar items are closely embedded, whereas dissimilar items are spaced further apart. This configuration simplifies the task for conventional clustering techniques. However, our proposed method focuses on the improvement of the clustering technique itself.

**Evaluation metric:** Unlike classifiers, where the type of classes and their predictions are kept intact, the prediction cluster indicators are permuted in clustering. Therefore, the clustering evaluation reduces to comparing two different types of sets, i.e., ground-truth labels set and cluster indicator. As a result, more than a single metric is required to assess all aspects of the clustering performance normalized mutual information (NMI) and accuracy (ACC) capture different aspects of the performance.

NMI in eq. (15) compares two sets using entropy as a criterion. It measures how much entropy is required to describe the second set using the entropy of the first set. Normalization scales the metric from zero to one.

$$\text{NMI}(\boldsymbol{X}, \boldsymbol{Y}) = \frac{\sum_{\mathbf{x} \in \boldsymbol{X}} \sum_{\mathbf{y} \in \boldsymbol{Y}} P(\mathbf{x}, \mathbf{y}) \log \frac{P(\mathbf{x}, \mathbf{y})}{P(\mathbf{x})P(\mathbf{y})}}{\max \left\{ -\sum_{\mathbf{x} \in \boldsymbol{X}} P(\mathbf{x}) \log P(\mathbf{x}), -\sum_{\mathbf{y} \in \boldsymbol{Y}} P(\mathbf{y}) \log P(\mathbf{y}) \right\}} \tag{15}$$

In contrast, accuracy (ACC in cf. eq. (16)) measures the pairwise exactness of the predicted clusters with the ground-truth clusters. This metric requires a bi-partite set alignment as in an unsupervised learning setup; the predicted cluster index does not correspond to the ground-truth label value at the individual data level. Therefore, the two sets ($\boldsymbol{X}$ and $\boldsymbol{Y}$ of size $n$) are aligned through the Kuhn-Munkres algorithm (KMA) (Kuhn, 1955) (*i.e.*, $\hat{\boldsymbol{Y}} = \text{KMA}(\boldsymbol{Y})$) upfront (see eq. (16)).

$$\text{ACC}[\boldsymbol{X}, \hat{\boldsymbol{Y}} = \text{KMA}(\boldsymbol{Y})] = \frac{\sum_{\mathbf{x} \in \boldsymbol{X}, \hat{\mathbf{y}} \in \hat{\boldsymbol{Y}}} \delta(\mathbf{x}, \hat{\mathbf{y}})}{n} \tag{16}$$

Table 2: ACC and NMI over the different datasets. The average performance is outside the brackets, while the highest performance from multiple runs is in the bracket for each method. MKKM and AASC results are an aggregation of multiple realizations.

| | | | | | ACC | | | | |
|---|---|---|---|---|---|---|---|---|---|
| **Data** | **SC** | **KKM** | **MKKM** | **RKKM** | **AASC** | **TLSC** | **SLKE-S** | **SLKE-R** | **Ours** |
| YALE | 40.53(49.42) | 38.97(47.12) | 45.70 | 39.71(48.09) | 40.64 | 45.35(55.85) | 38.89(61.82) | 51.28(66.24) | **52.62** |
| JAFFE | 54.03(74.88) | 67.09(74.39) | 74.55 | 67.89(75.61) | 30.35 | 86.64(99.83) | 70.77(96.71) | **90.89**(99.85) | 86.38 |
| ORL | 46.65(58.96) | 45.93(53.53) | 47.51 | 46.88(54.96) | 27.20 | 50.50(62.35) | 45.32(77.00) | 59.00(74.75) | **67.75** |
| COIL20 | 43.65(67.60) | 50.74(59.49) | 54.82 | 51.89(61.64) | 34.87 | 38.03(72.71) | 56.83(75.42) | 65.55(84.03) | **69.79** |
| BA | 26.25(31.07) | 33.66(41.20) | 40.52 | 34.35(42.17) | 27.07 | 39.50(47.72) | 36.35(50.74) | 35.79(44.37) | **43.87** |
| TR11 | 43.32(50.98) | 44.65(51.91) | 50.13 | 45.04(53.03) | 47.15 | 54.79(71.26) | 46.87(69.32) | **55.07**(74.64) | 50.96 |
| TR41 | 44.80(63.52) | 46.34(55.64) | **56.10** | 46.80(56.76) | 45.90 | 43.18(65.60) | 47.91(71.19) | 53.51(74.37) | 55.69 |
| TR45 | 45.96(57.39) | 45.58(58.79) | 58.46 | 45.69(58.13) | 52.64 | 53.38(74.02) | 50.59(78.55) | 58.37(79.89) | **63.32** |
| | | | | | NMI | | | | |
| YALE | 44.79(52.92) | 42.07(51.34) | 50.06 | 42.87(52.29) | 46.83 | 45.07(56.50) | 40.38(59.47) | 52.87(64.29) | **62.05** |
| JAFFE | 59.35(82.08) | 71.48(80.13) | 79.79 | 74.01(83.47) | 27.22 | 84.67(99.35) | 60.83(94.80) | 81.56(99.49) | **93.90** |
| ORL | 66.74(75.16) | 63.36(73.43) | 68.86 | 63.91(74.23) | 43.77 | 63.55(78.96) | 58.84(86.35) | 75.34(85.15) | **86.30** |
| COIL20 | 54.34(80.98) | 63.57(74.05) | 70.64 | 63.70(74.63) | 41.87 | 73.26(82.20) | 65.40(80.61) | 73.53(91.25) | **85.32** |
| BA | 40.09(50.76) | 46.49(57.25) | 56.88 | 46.91(57.82) | 42.34 | 52.17(63.04) | 55.06(63.58) | 50.11(56.78) | **60.51** |
| TR11 | 31.39(43.11) | 33.22(48.88) | 44.56 | 33.48(49.69) | 39.39 | 37.58(58.60) | 30.56(67.63) | 45.39(70.93) | **48.68** |
| TR41 | 36.60(61.32) | 40.37(59.88) | **57.75** | 40.86(60.77) | 43.05 | 43.18(65.50) | 34.82(70.89) | 47.45(68.50) | 51.15 |
| TR45 | 33.22(48.03) | 38.69(57.87) | **56.17** | 38.96(57.86) | 41.94 | 44.36(74.24) | 38.04(72.50) | 50.37(78.12) | 55.93 |

Table 3: ACC and NMI over the different datasets. Comping methods have the average performance while ours is only one estimation.

| | | | | | ACC | | | | | | |
|---|---|---|---|---|---|---|---|---|---|---|---|
| **Data** | **SelfT** | **JSESR** | **LRR** | **SSC** | **CAN** | **PCAN** | **SWCAN** | **LAPIN** | **DOGS** | **JGSED** | **Ours** |
| YALE | 47.65 | 30.18 | 26.67 | 24.48 | 24.24 | 25.45 | 26.67 | 30.45 | 30.93 | 32.12 | **52.62** |
| JAFFE | 54.28 | 27.83 | 29.72 | 29.72 | 28.77 | 28.30 | 30.19 | 30.68 | 31.79 | 32.08 | **86.38** |
| COIL20 | 38.17 | 81.37 | 73.40 | 76.18 | 83.54 | 83.33 | 79.03 | 81.96 | 80.90 | **82.64** | 69.79 |
| BA | 23.67 | 46.59 | 44.59 | 45.87 | 42.24 | 43.59 | 46.08 | 46.83 | 38.53 | **47.86** | 43.87 |
| | | | | | NMI | | | | | | |
| YALE | 59.64 | 33.48 | 27.53 | 25.69 | 20.71 | 23.54 | 24.05 | 32.98 | 33.42 | 33.78 | **62.05** |
| JAFFE | 69.86 | 9.97 | 16.62 | 14.92 | 13.33 | 12.88 | 15.33 | 15.91 | 18.06 | 17.59 | **93.90** |
| COIL20 | 63.15 | 89.25 | 82.90 | 90.29 | 91.09 | 89.08 | 89.19 | 90.03 | 88.96 | **91.17** | 85.32 |
| BA | 38.96 | 60.83 | 57.93 | **62.50** | 54.24 | 49.55 | 57.98 | 59.04 | 49.82 | 60.64 | 60.51 |

**Results:** One can notice that the model outperforms the previously reported method on an NMI metric by up to 10% (see table 2). This indicates that the entropy of the cluster indexes generated by our similarity matrix is closer to the one actual class index for the data at hand. Furthermore, the amount of improvement (in the NMI term) relative to vanilla SC is quite considerable. This suggests that the proposed method is more effective than the baseline and SOTA methods at identifying gaps in the data topology. NMI makes a more global comparison between true labels and predictions; therefore, we compute clustering accuracy as a complementary metric to provide the comparison at the individual level. The accuracy has improved for nearly all previously proposed methods. Nevertheless, it is important to note that the evaluation of competing methods is based on average accuracy, which means a less accurate clustering assignment is just as possible. As in the case of NMI, our proposed method significantly improves accuracy compared to the SC baseline across all test image datasets. Similarly, optimizing the kernel bandwidth $\sigma$ follows a

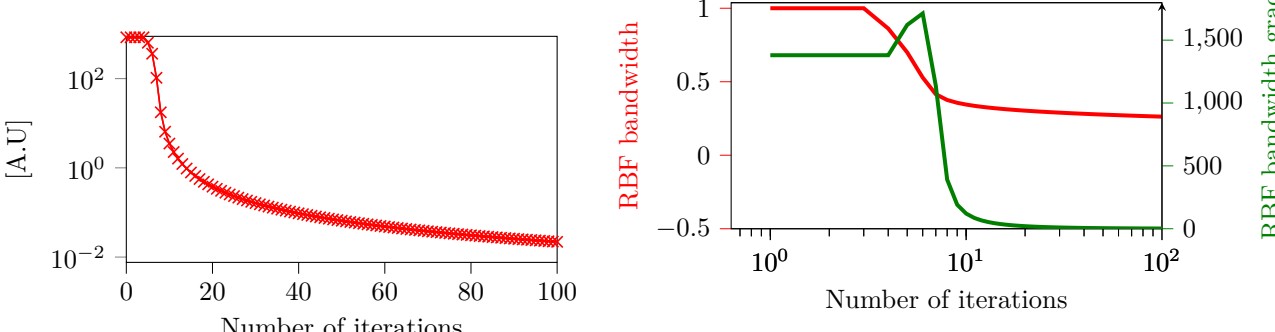

(a) Loss function (loss $= \sum_{i=1}^{K} \lambda_i$) over during training.    (b) RBF bandwidth $\sigma$ and its gradient during training.

Figure 5: Optimization of the $\sigma$ on the JAFFE dataset in fig. 5b. Cumulative magnitude reduction of the fist K=10 eigenvalues (loss $= \sum_{i=1}^{K} \lambda_i$) over the iterations for the JAFFE dataset in fig. 5a.

similar trajectory as in the benchmark dataset. The method progressively tries to reach an optimal value and decreases the gradient (see figs. 5b, 16b, 17b, 18b, 19b, 20b, 21b and 22b in Appendix). Likewise, the cumulative magnitude of the first $K$ eigenvalues progresses towards smaller values as in the case of the benchmark data experiments (see figs. 5a, 16a, 17a, 18a, 19a, 20a, 21a and 22a in Appendix). This behaviour indicates the formation of the clusters as the $\sigma$ progresses towards more optimal values.

## 5    Conclusion

Similarity learning for SC operates under the optimization objective that maximizes the linear separation for K clusters when using the first K eigenvectors of the Laplacian matrix. One can search for the optimal parameter for the RBF kernel as a similarity function through the proposed optimization objective. Alternatively, arbitrarily selecting the RBF bandwidth risks choosing a suboptimal value that might either underemphasize the distances, resulting in indistinguishable clusters, or overemphasize them, leading to individual data being isolated as its own cluster.

The goal of the proposed method is to function without labels, ideally in a fully unsupervised manner that eliminates the dependence on hyperparameters. When working without labeled data, validating any hyperparameters directly through the dataset or the objective function is impossible. Moreover, choosing suboptimal hyperparameter values can negatively impact the effectiveness of clustering in subsequent tasks. Consequently, our work introduces a framework for similarity learning in SC that avoids the introduction of any hyperparameters.

We showcase our method's ability to separate highly non-convex shapes and project the dataset into a linearly separable topology on a series of benchmark datasets. The proposed method is validated on image and text datasets and compared with vanilla SC as a baseline and recent similarity learning models. Our proposed method outperforms competing approaches in both normalized mutual information and accuracy. Notice that, all alternative models' performance strictly depends on their hyperparameter settings. *Since these hyperparameters are not validated in an unsupervised setting, the optimal performance of the competing methods is never guaranteed.*

As a future work, it should be noted that SC itself inherently requires the predefined specification of the number of clusters $K$ within the dataset. Looking forward, an area for further research would be incorporating methods for autonomously estimating the number of clusters. A promising approach is to identify density peaks within the data's topological structure, an idea inspired by the technique presented in the study by Rodriguez & Laio (2014). Adopting such a strategy could enhance the unsupervised aspect of SC, expanding it from similarity learning to include autonomous cluster number determination.

The proposed method is constrained to the RBF kernel, necessitating precise pairwise image alignment, which is impractical for real-world image datasets. To increase applicability, the approach should incorporate deep learning models, which produce translation-invariant embeddings, thus reducing the need for exact pairwise image alignment.

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

## A  Non-negativity of the gradient of the loss

**Theorem 1.** *The gradient of loss function is non-negative:* $\frac{\partial loss(\sigma)}{\partial \sigma} \geq 0, \forall \sigma \in \mathbb{R}_{\geq 0}$.

*Proof.* This can be easily shown by proving that:

$$\frac{\partial \text{loss}(\sigma)}{\partial \sigma} = \sum_{i=1}^{K} \frac{\partial \{\lambda_i(\sigma)\}}{\partial \sigma} \geq 0 \tag{17}$$

Differentiating the eigenvalues of a symmetric matrix (i.e., $d\lambda = \boldsymbol{f} d\boldsymbol{L} \boldsymbol{f}^T$ (Petersen & Pedersen, 2008))

$$\frac{\partial \text{loss}(\sigma)}{\partial \sigma} = \sum_{i=1}^{K} \frac{\partial \{\lambda_i(\sigma)\}}{\partial \sigma} = \sum_{i=1}^{K} f_i \frac{\partial \boldsymbol{L}(\sigma)}{\partial \sigma} f_i^T \geq 0, \forall \sigma \in \mathbb{R}_{\geq 0} \tag{18}$$

Given that $\boldsymbol{L}(\sigma)$ is itself symmetric, its differentiation w.r.t $\sigma \in \mathbb{R}_{\geq 0}$ is also another real symmetric Laplacian matrix (see lemma 1). Since any real symmetric Laplacian matrix is a positive semi-definite (PSD) (von Luxburg, 2007), its Rayleigh quotient is always positive (see (Strang, 2006)). Hence, even for eigenvector $\boldsymbol{f}$ , the result is always non-negative $f_i \frac{\partial \boldsymbol{L}(\sigma)}{\partial \sigma} f_i^T \geq 0, \forall i \in \{1, .., K\}$ resulting in $\sum_{i=1}^{K} f_i \frac{\partial \boldsymbol{L}(\sigma)}{\partial \sigma} f_i^T \geq 0, \forall \boldsymbol{f} \in \mathbb{R}^N$. $\qquad \square$

**Lemma 1.** *The derivative of the Laplacian matrix $\left(\frac{\partial \boldsymbol{L}(\sigma)}{\partial \sigma}\right)$ is positive semi-definite (PSD) $\forall \sigma \in \mathbb{R}_{\geq 0}$.*

*Proof.* Initially once can easily observe that because $\boldsymbol{L}(\sigma) = \boldsymbol{L}^T(\sigma)$ is symmetric its derivative is also symmetric $\frac{\partial \boldsymbol{L}(\sigma)}{\partial \sigma} = \frac{\partial \boldsymbol{L}^T(\sigma)}{\partial \sigma}$ (see corollary 1). Since the domain of $\sigma \in \Sigma \subset \mathbb{R}_{\geq 0}$, it follows that the derivative of $\boldsymbol{L}$ constitutes an alternate Laplacian matrix, (see lemma 2).

Given the new Laplacian matrix (i.e., $\frac{\partial \boldsymbol{L}(\sigma)}{\partial \sigma} = \frac{\partial \boldsymbol{L}^T(\sigma)}{\partial \sigma}$) is symmetric, its eigenvalues are real values (Strang, 2006). As any other Laplacian matrix, $\frac{\partial \boldsymbol{L}(\sigma)}{\partial \sigma}$ can be written as a product of incidence matrix (i.e., $\boldsymbol{M}$) as $\frac{\partial \boldsymbol{L}(\sigma)}{\partial \sigma} = \boldsymbol{M}^T \boldsymbol{M}$. Whenever the eigenvalues are real numbers, one can safely say that $\forall \boldsymbol{x}, \boldsymbol{x}^T \frac{\partial \boldsymbol{L}(\sigma)}{\partial \sigma} \boldsymbol{x} = \rho \in \mathbb{R}_{\geq 0}$. Reusing $\frac{\partial \boldsymbol{L}(\sigma)}{\partial \sigma} = \boldsymbol{M}^T \boldsymbol{M}$ into $\boldsymbol{x}^T \frac{\partial \boldsymbol{L}(\sigma)}{\partial \sigma} \boldsymbol{x} = \boldsymbol{x}^T \boldsymbol{M}^T \boldsymbol{M} x = ||\boldsymbol{M} \boldsymbol{x}||^2 = \rho \to \rho \geq 0$. Hence, the new Laplacian matrix $\frac{\partial \boldsymbol{L}(\sigma)}{\partial \sigma}$ is PSD. $\qquad \square$

**Lemma 2.** *The derivative of Laplacian matrix $\left(\frac{\partial \boldsymbol{L}(\sigma)}{\partial \sigma}\right)$ is another Laplacian matrix $\forall \sigma \in \mathbb{R}_{\geq 0}$.*

*Proof.* To prove that the derivative of the Laplacian matrix $\left(\frac{\partial \boldsymbol{L}(\sigma)}{\partial \sigma}\right)$ is another Laplacian matrix, one must initially demonstrate that each diagonal entry is the negative sum of the non-diagonal entries in its corresponding row (or equivalently, its column), owing to the symmetry of the matrix. Since the matrix is symmetric, we choose row-wise, equivalent to column-wise, and vice versa. Furthermore, it suffices to show for just one row; without the loss of generality, it is equivalent to the rest of the rows.

$$\frac{\partial L_{j,j}(\sigma)}{\partial \sigma} = \frac{\partial \sum_{i=0}^{N-1} A_{j,i}(\sigma)}{\partial \sigma} = \sum_{i=0}^{N-1} \frac{\partial A_{i,j}(\sigma)}{\partial \sigma} \tag{19}$$

Given that $\frac{dK(\sigma^2)}{d\sigma} = 2\sigma \frac{d(x,y)}{(\sigma^2)^2} e^{\frac{-d(x,y)}{\sigma^2}} > 0, \forall \sigma, d(x,y) \in \mathbb{R}_{\geq 0}$. This results in two important implications:

- Firstly, the diagonal values for $\frac{\partial L_{i,i}(\sigma)}{\partial \sigma}$ are always positive since $\frac{\partial A_{i,j|i\neq j}(\sigma)}{\partial \sigma} > 0, \forall i \in [0, N-1], \forall \sigma, d(x,y) \in \mathbb{R}_{\geq 0}$.

- Secondly, off diagonal of entries of $\frac{\partial L_{i,j|i\neq j}(\sigma)}{\partial \sigma} = -\frac{\partial K(\sigma)}{\partial \sigma} < 0, \forall \sigma, d(x,y) \in \mathbb{R}_{\geq 0}$

Hence, together with eq. (19), one can conclude that the new matrix is a symmetric Laplacian matrix (see eq. (20)).

$$\frac{\partial \boldsymbol{L}(\sigma)}{\partial \sigma} = \frac{\partial \boldsymbol{L}^T(\sigma)}{\partial \sigma} \tag{20}$$

$\square$

**Corollary 1.** *The derivative of Laplacian matrix $\boldsymbol{L}$, $\frac{\partial \boldsymbol{L}(\sigma)}{\partial \sigma}$ is a symmetric matrix.*

*Proof.* It is crucial to compute the $\frac{\partial \boldsymbol{L}(\sigma)}{\partial \sigma}$ using the RBF as a kernel to calculate the second derivative of the RBF itself first.

Using the composition $\frac{d}{dx}[f(g(x))] = \frac{d}{dg}[f(g)]\frac{d}{dx}[g(x)]$ the derivative could be simplified to:

$$\frac{dK(\sigma^2)}{d\sigma} = \frac{dK(\sigma^2)}{d\sigma^2}\frac{d\sigma^2}{d\sigma} = 2\sigma\frac{dK(\sigma^2)}{d\sigma^2} \tag{21}$$

where the first derivative of the kernel is:

$$\frac{dK(\sigma^2)}{d\sigma^2} = e^{\frac{-d(x,y)}{\sigma^2}}\frac{d(x,y)}{(\sigma^2)^2} \tag{22}$$

Combining eq. (22) and eq. (21) results into eq. (23).

$$\frac{dK(\sigma^2)}{d\sigma} = 2\sigma\frac{d(x,y)}{(\sigma^2)^2}e^{\frac{-d(x,y)}{\sigma^2}} \tag{23}$$

As a result of differentiation, the RBF kernel transforms into another different similarity metric eq. (24). The new similarity metric is not a kernel since it contains the Euclidean distance, which is not a kernel function.

$$\frac{dK(\sigma^2)}{d\sigma}(x,y) = \frac{dK(\sigma^2)}{d\sigma}(y,x) \tag{24}$$

As a result of symmetricity(see eq. (25)), the new matrix resulting from the differentiation of the Laplacian would still be another symmetricity matrix.

$$\frac{\partial \boldsymbol{L}(\sigma)}{(\partial \sigma)} = \left\{\frac{\partial \boldsymbol{L}(\sigma)}{\partial \sigma}\right\}^T = \frac{\partial \boldsymbol{L}^T(\sigma)}{\partial \sigma} \tag{25}$$

$\square$

# B  Experiments on benchmark dataset

Herein, we present a series of sanity checks on popular benchmark clustering datasets (see table 4). The purpose of these experimentations is to showcase the method's capability to separate clusters under different types of convexity. Apart from different types of separation, the proposed method also handles an asymmetric number of data per cluster as in fig. 9. For each tested dataset, we present the final eigen-embeddings, the loss (loss $= \sum_{i=1}^{K} \lambda_i$) over iterations together with the trajectory of the RBF bandwidth and its gradient. A visual representation of the eigen-embeddings illustrates how the method facilitates the linear separation of individual clusters.

Moreover, we have showcased the robustness of our analysis by introducing two outliers to these datasets (refer to figs. 11 to 15). This addition serves to test the resilience of our method against anomalies. In our visual representations, we have chosen to exclude the first two eigenvectors, since their primary function is to separate the outliers from the rest of the data.

Table 4: Description of the benchmark datasets.

| Dataset | Size | Nr of Dimensions | Nr of Classes |
|---|---|---|---|
| Two spirals (see fig. 3) | 500 | 3 | 2 |
| Two rings (see fig. 6) | 500 | 3 | 2 |
| Three spirals (see fig. 7) | 312 | 2 | 3 |
| Two circles (see fig. 8) | 500 | 2 | 2 |
| Two (asymmetric) half moons (see fig. 8) | 373 | 2 | 2 |
| Two (symmetric) half moons (see fig. 9) | 500 | 2 | 2 |

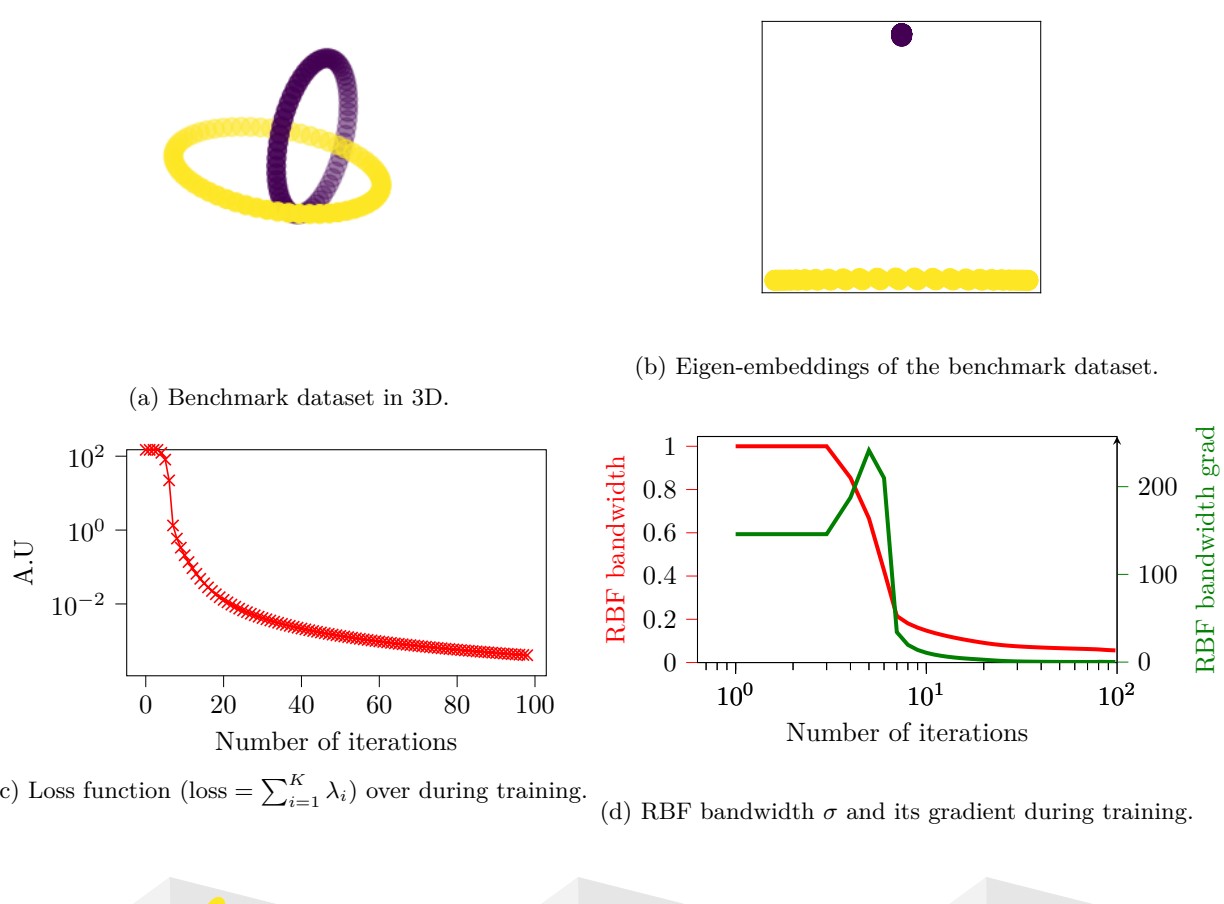

(a) Benchmark dataset in 3D.

(b) Eigen-embeddings of the benchmark dataset.

(c) Loss function (loss $= \sum_{i=1}^{K} \lambda_i$) over during training.

(d) RBF bandwidth $\sigma$ and its gradient during training.

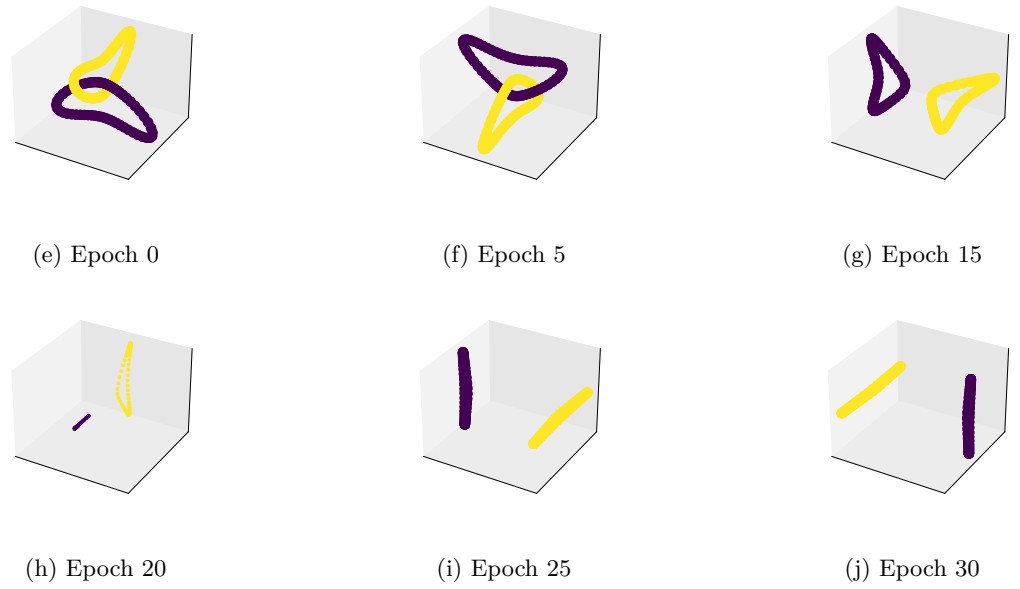

(e) Epoch 0     (f) Epoch 5     (g) Epoch 15

(h) Epoch 20     (i) Epoch 25     (j) Epoch 30

Figure 6: Benchmark dataset two rings in 3D (see fig. 6a). Despite the shape not being convex, the method learns a linear separation of these two rings (see fig. 6b). The loss values (loss $= \sum_{i=1}^{K} \lambda_i$) decrease consistently with the number of iterations (see fig. 6c). Optimizing kernel bandwidth $\sigma$ using the gradient descent stabilizes as the $\sigma$ gradient diminishes (see fig. 6d). A snapshot of the optimized trajectory is in figs. 6e to 6j.

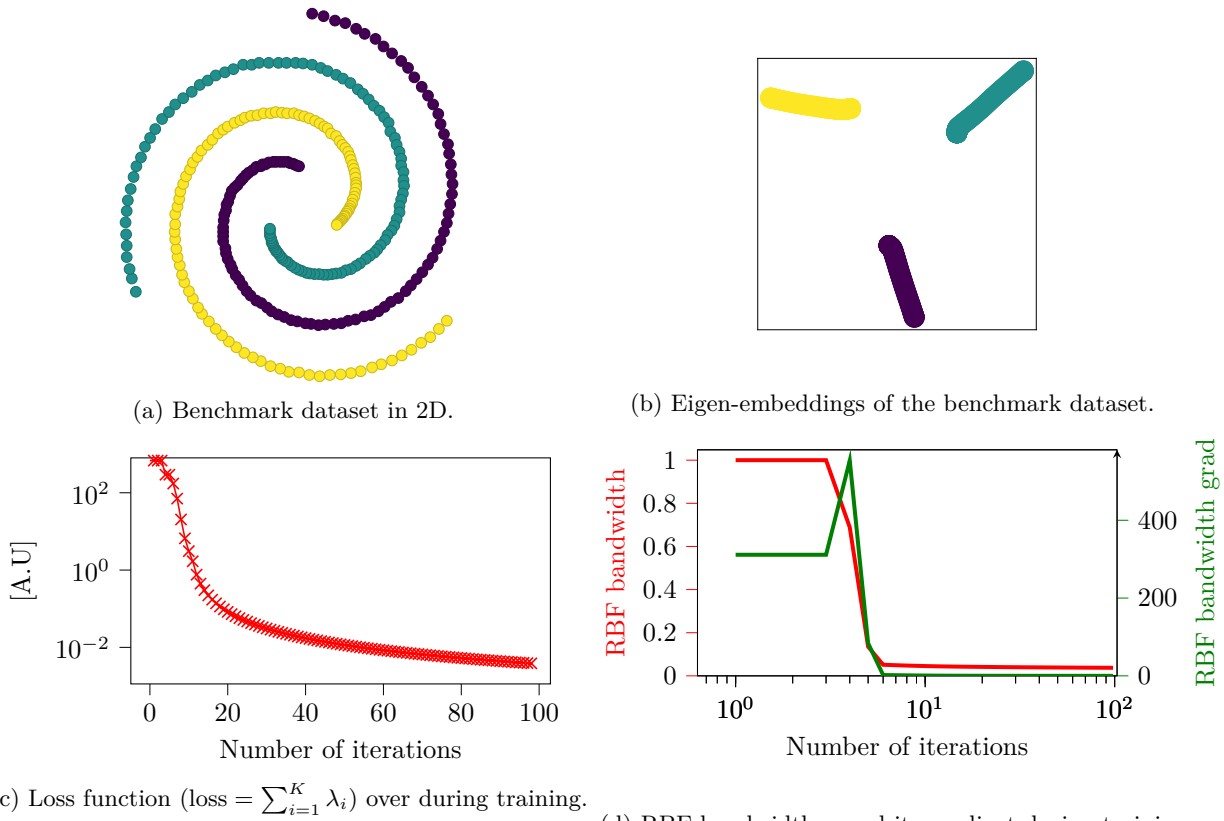

(a) Benchmark dataset in 2D.

(b) Eigen-embeddings of the benchmark dataset.

(c) Loss function (loss $= \sum_{i=1}^{K} \lambda_i$) over during training.

(d) RBF bandwidth $\sigma$ and its gradient during training.

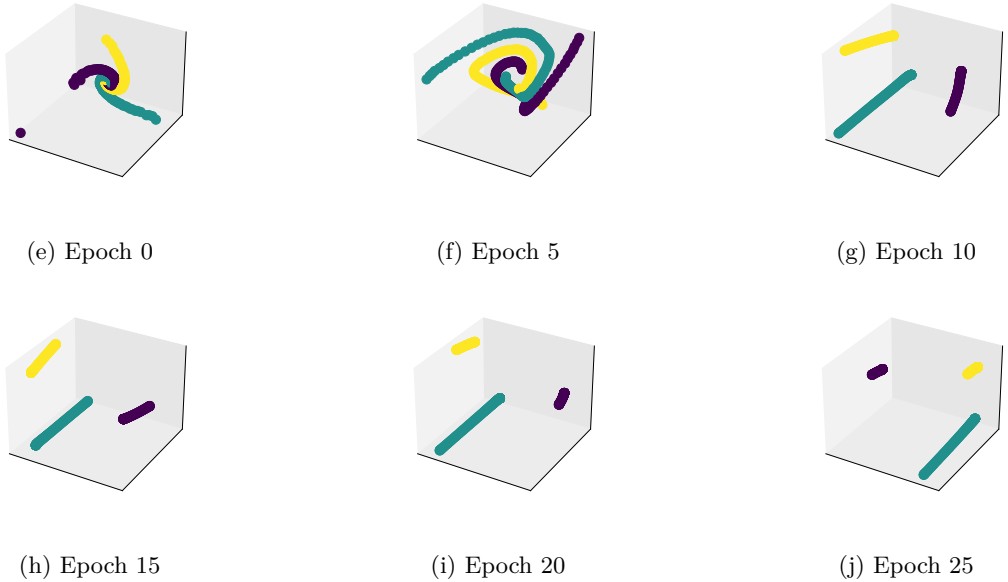

(e) Epoch 0

(f) Epoch 5

(g) Epoch 10

(h) Epoch 15

(i) Epoch 20

(j) Epoch 25

Figure 7: Benchmark dataset three spirals in 2D (see fig. 7a). Despite the shape not being convex, the method learns a linear separation of these three spirals (see fig. 7b). The loss values (loss $= \sum_{i=1}^{K} \lambda_i$) decrease consistently with the number of iterations (see fig. 7c). Optimizing kernel bandwidth $\sigma$ using the gradient descent stabilizes as the $\sigma$ gradient diminishes (see fig. 7d). A snapshot of the optimized trajectory is in figs. 7e to 7i.

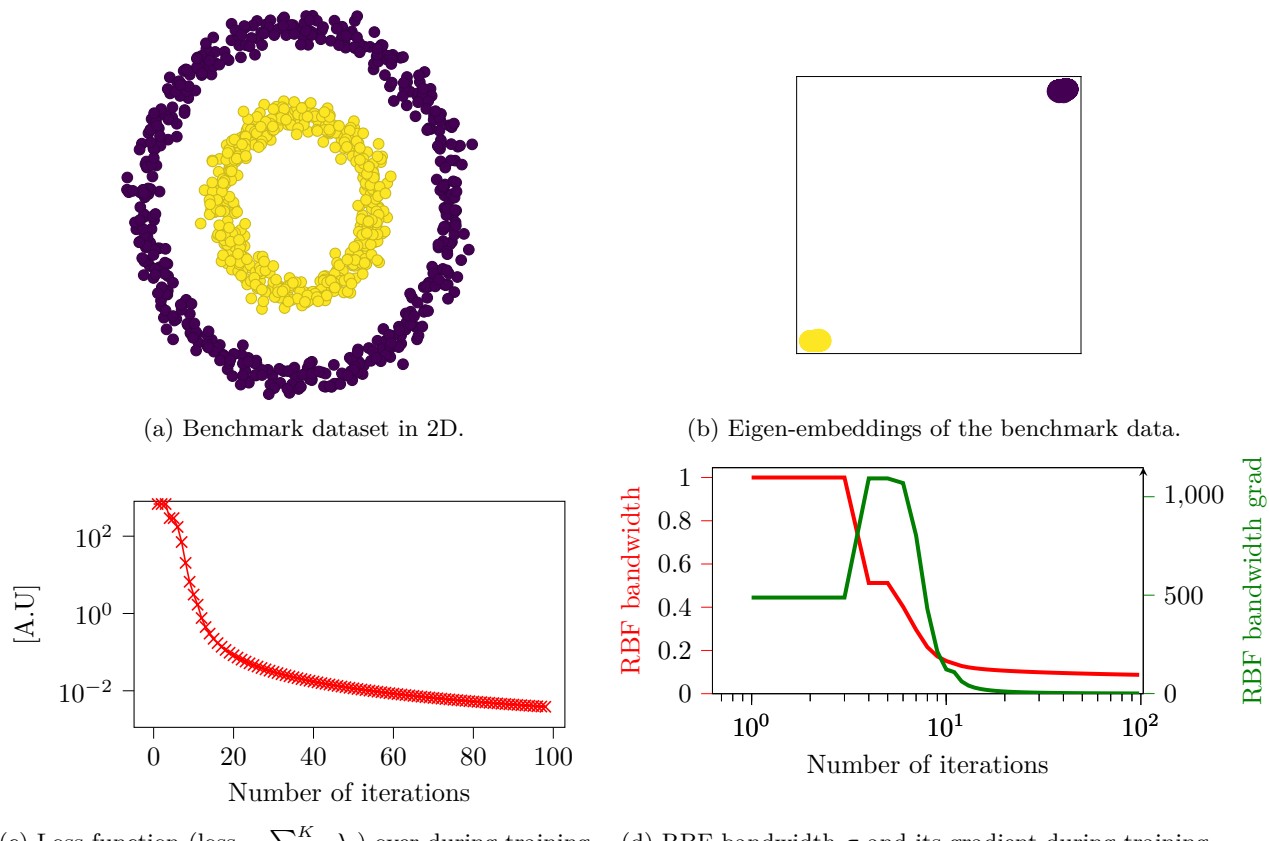

(a) Benchmark dataset in 2D.

(b) Eigen-embeddings of the benchmark data.

(c) Loss function (loss $= \sum_{i=1}^{K} \lambda_i$) over during training.

(d) RBF bandwidth $\sigma$ and its gradient during training.

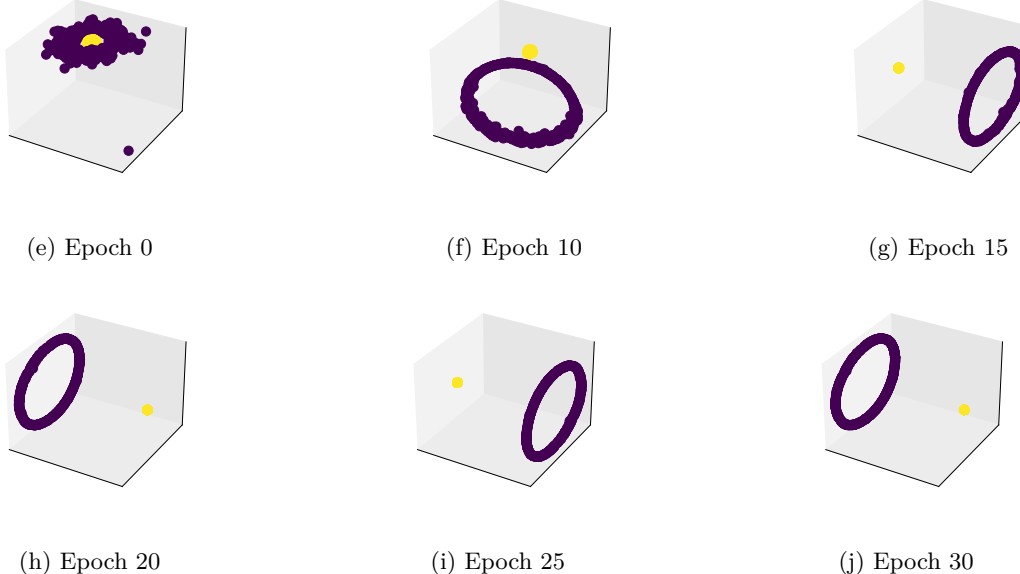

(e) Epoch 0

(f) Epoch 10

(g) Epoch 15

(h) Epoch 20

(i) Epoch 25

(j) Epoch 30

Figure 8: Benchmark dataset two rings in 2D (see fig. 8a). Despite the shape not being convex, the method learns a linear separation of these two rings (see fig. 8b). The loss values (loss $= \sum_{i=1}^{K} \lambda_i$) decrease consistently with the number of iterations (see fig. 8c). Optimizing kernel bandwidth $\sigma$ using the gradient descent stabilizes as the $\sigma$ gradient diminishes (see fig. 8d). A snapshot of the optimized trajectory is in figs. 8e to 8j.

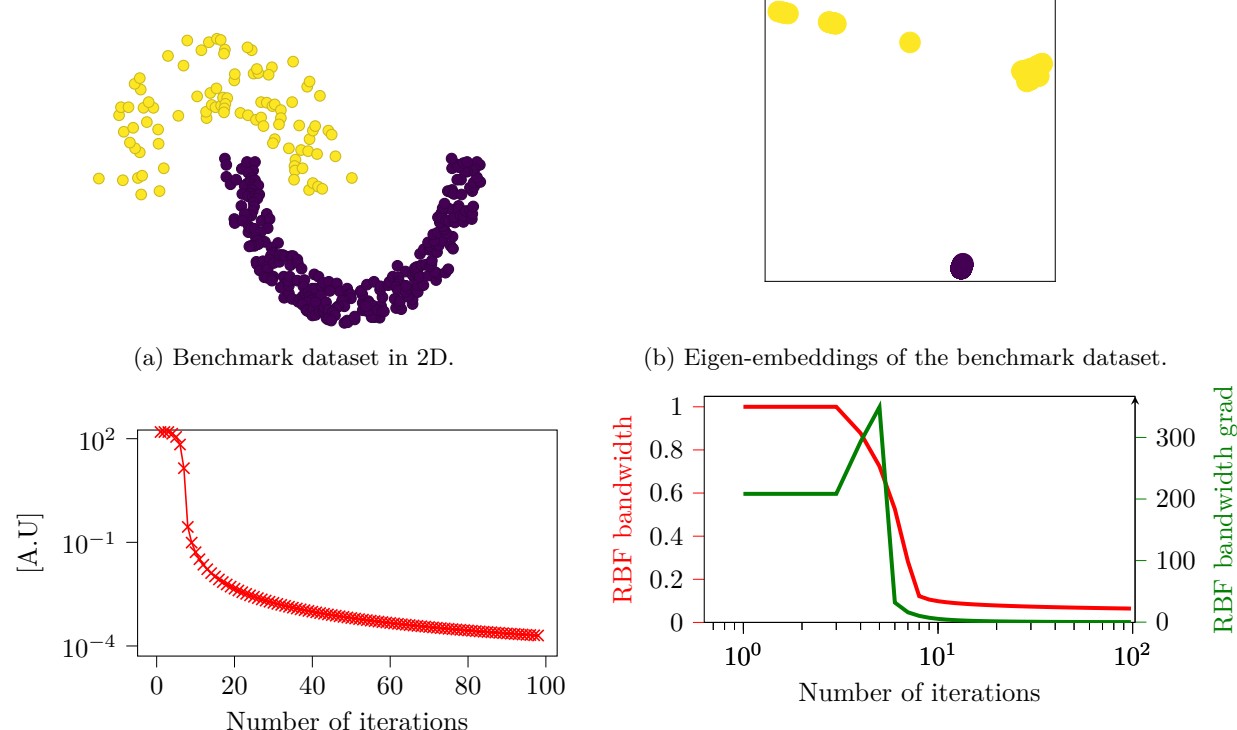

(a) Benchmark dataset in 2D.

(b) Eigen-embeddings of the benchmark dataset.

(c) Loss function (loss $= \sum_{i=1}^{K} \lambda_i$) over during training.

(d) RBF bandwidth $\sigma$ and its gradient during training.

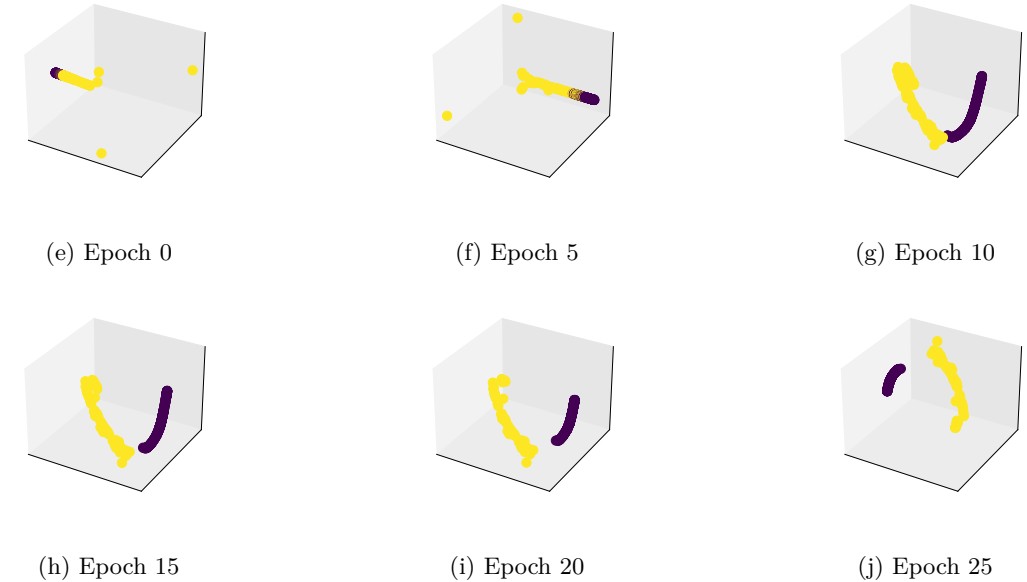

(e) Epoch 0

(f) Epoch 5

(g) Epoch 10

(h) Epoch 15

(i) Epoch 20

(j) Epoch 25

Figure 9: Benchmark dataset two (asymmetric) half moons in 2D (see fig. 9a). Despite the shape not being convex, the method learns a linear separation of these two half moons (see fig. 9b). The loss values (loss $= \sum_{i=1}^{K} \lambda_i$) decrease consistently with the number of iterations (see fig. 9c). Optimizing kernel bandwidth $\sigma$ using gradient descent stabilizes as the $\sigma$ gradient diminishes (see fig. 9d). A snapshot of the optimized trajectory is in figs. 9e to 9j.

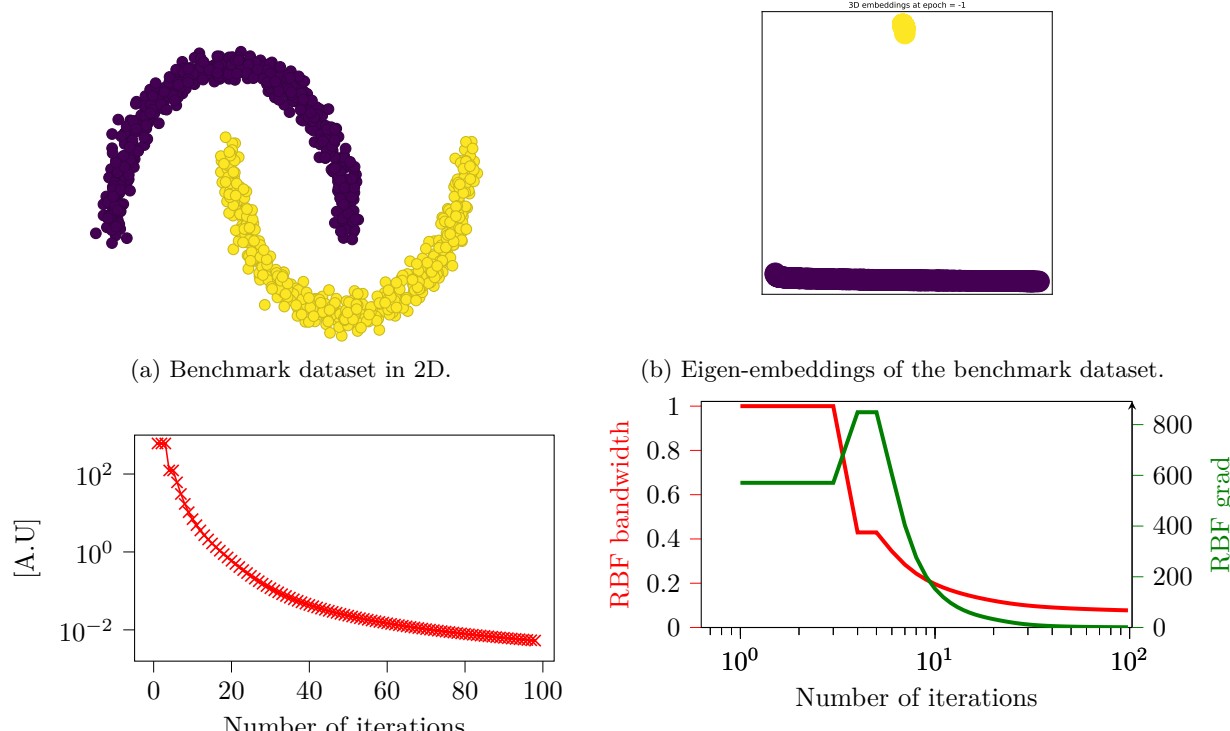

(a) Benchmark dataset in 2D.

(b) Eigen-embeddings of the benchmark dataset.

(c) Loss function (loss $= \sum_{i=1}^{K} \lambda_i$) over during training.

(d) RBF bandwidth $\sigma$ and its gradient during training.

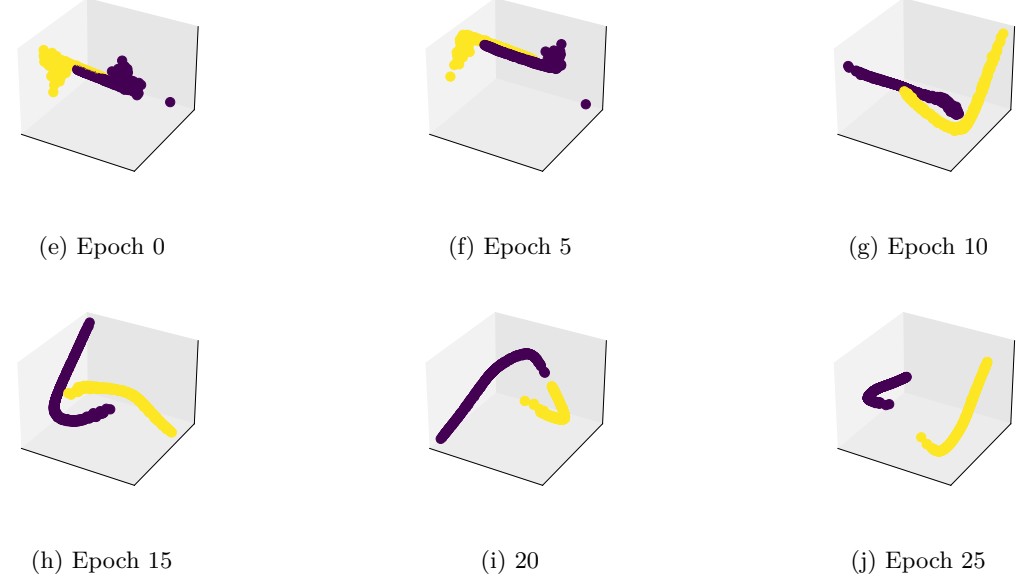

(e) Epoch 0

(f) Epoch 5

(g) Epoch 10

(h) Epoch 15

(i) 20

(j) Epoch 25

Figure 10: Benchmark dataset two moons in 2D (see fig. 10a). Despite the shape not being convex, the method learns a linear separation of these two moons (see fig. 10b). The loss values (loss $= \sum_{i=1}^{K} \lambda_i$) decrease consistently with the number of iterations(see fig. 10c). Optimizing kernel bandwidth $\sigma$ using gradient descent stabilizes as the $\sigma$ gradient diminishes (see fig. 10d). A snapshot of the optimized trajectory is in figs. 10e to 10j.

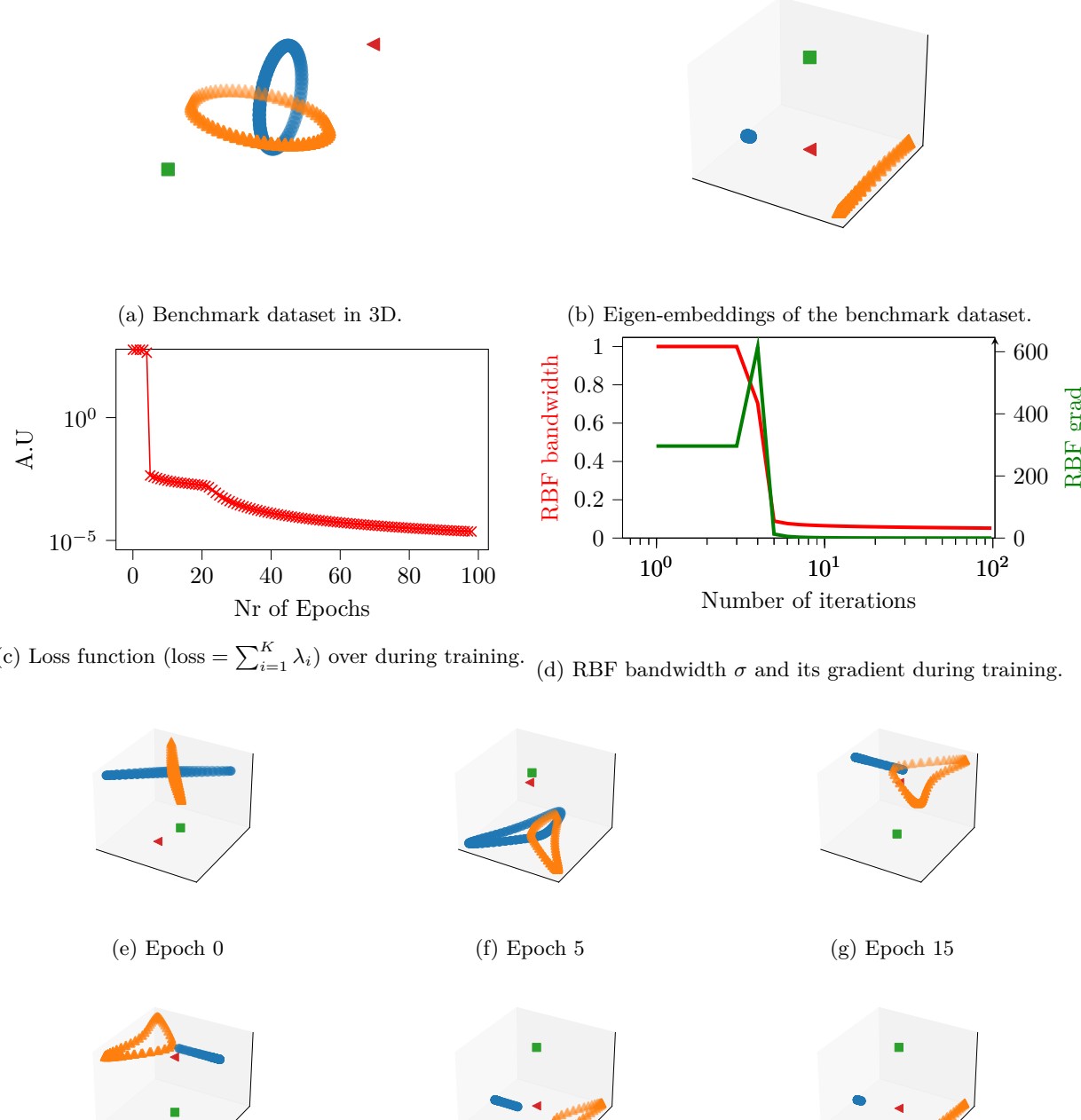

(a) Benchmark dataset in 3D.

(b) Eigen-embeddings of the benchmark dataset.

(c) Loss function (loss $= \sum_{i=1}^{K} \lambda_i$) over during training.

(d) RBF bandwidth $\sigma$ and its gradient during training.

(e) Epoch 0

(f) Epoch 5

(g) Epoch 15

(h) Epoch 20

(i) Epoch 25

(j) Epoch 25

Figure 11: Benchmark dataset two rings in 3D (see fig. 11a). Despite the presence of the outliers and the shape not being convex, the method learns a linear separation of these two rings (see fig. 11b). The loss values (loss $= \sum_{i=1}^{K} \lambda_i$) decrease consistently with the number of iterations (see fig. 11c). Optimizing kernel bandwidth $\sigma$ using the gradient descent stabilizes as the $\sigma$ gradient diminishes (see fig. 11d). A snapshot of the optimized trajectory is in figs. 11e to 11j, where the first two eigenvectors (corresponding to the outliers) are omitted and the subsequent ones are visualized instead.

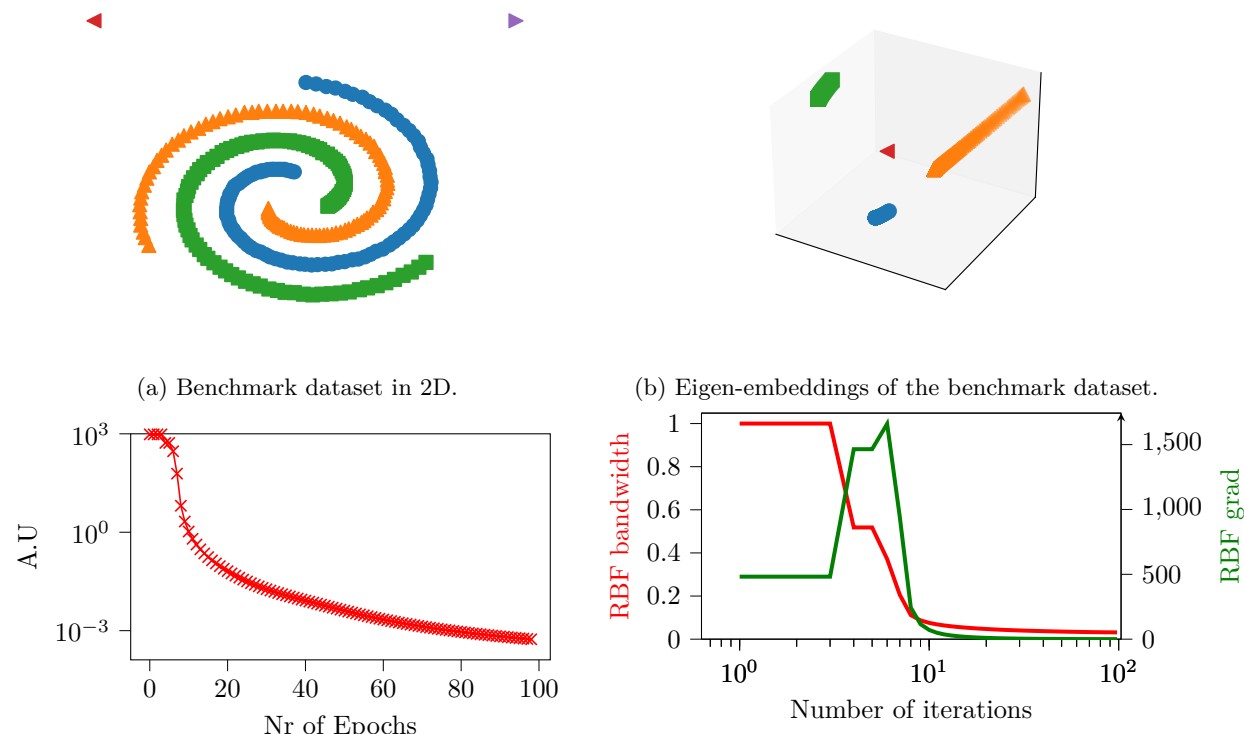

(a) Benchmark dataset in 2D.

(b) Eigen-embeddings of the benchmark dataset.

(c) Loss function (loss $= \sum_{i=1}^{K} \lambda_i$) over during training.

(d) RBF bandwidth $\sigma$ and its gradient during training.

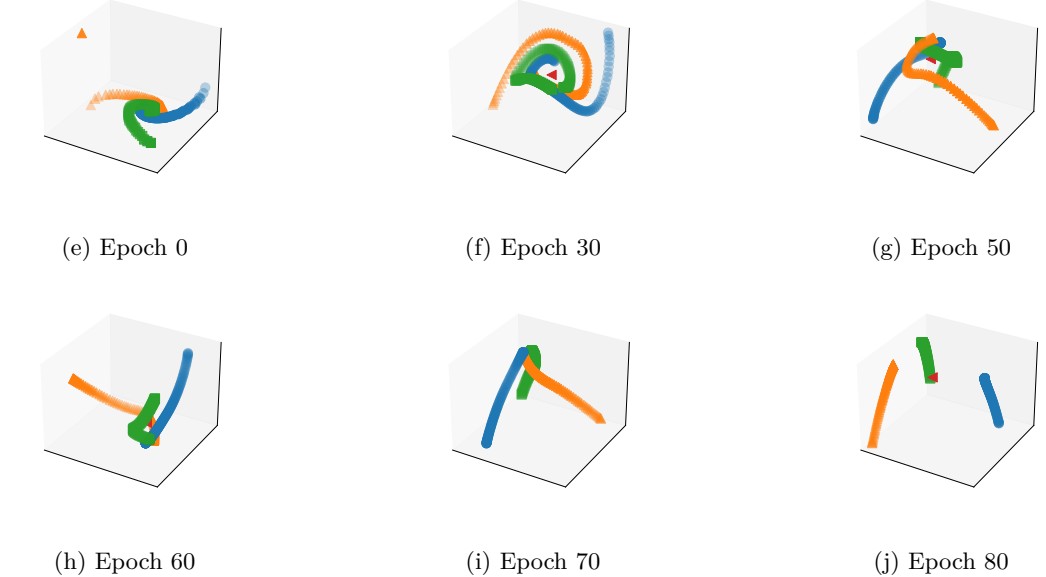

(e) Epoch 0

(f) Epoch 30

(g) Epoch 50

(h) Epoch 60

(i) Epoch 70

(j) Epoch 80

Figure 12: Benchmark dataset three spirals in 2D (see fig. 12a). Despite the presence of the outliers and the shape not being convex, the method learns a linear separation of these three spirals (see fig. 12b). The loss values (loss $= \sum_{i=1}^{K} \lambda_i$) decrease consistently with the number of iterations (see fig. 12c). Optimizing kernel bandwidth $\sigma$ using the gradient descent stabilizes as the $\sigma$ gradient diminishes (see fig. 12d). A snapshot of the optimized trajectory is in figs. 12e to 12i, where the first two eigenvectors (corresponding to the outliers) are omitted and the subsequent ones are visualized instead.

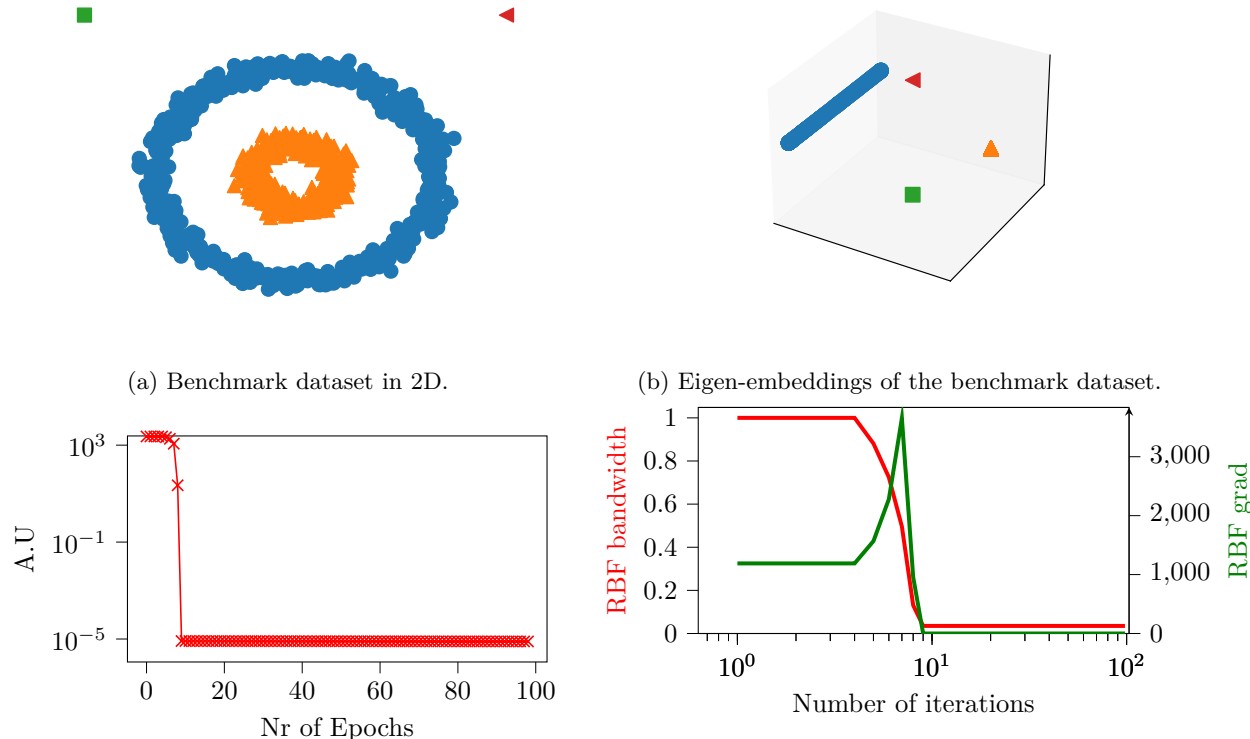

(a) Benchmark dataset in 2D.

(b) Eigen-embeddings of the benchmark dataset.

(c) Loss function (loss $= \sum_{i=1}^{K} \lambda_i$) over during training.

(d) RBF bandwidth $\sigma$ and its gradient during training.

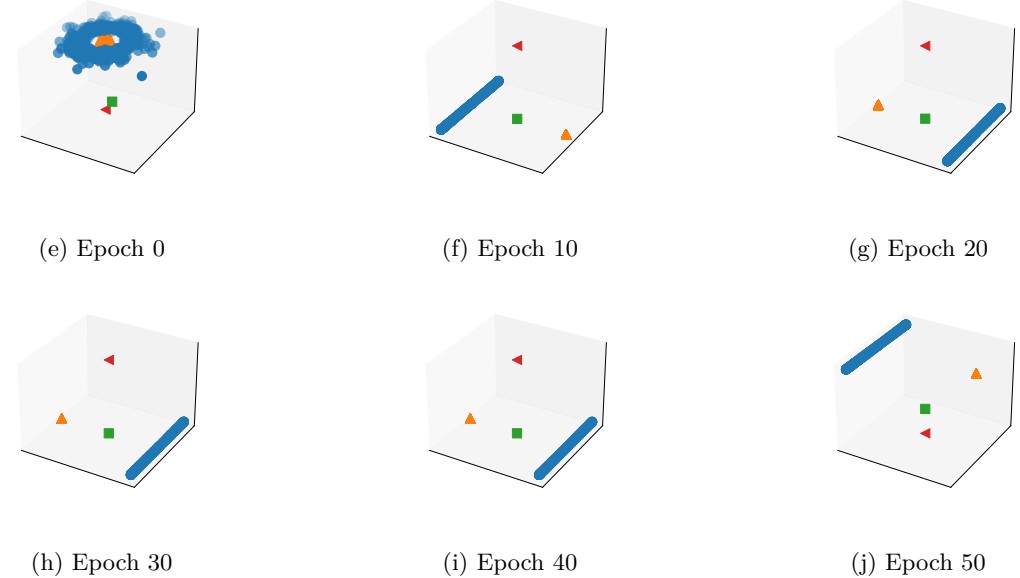

(e) Epoch 0

(f) Epoch 10

(g) Epoch 20

(h) Epoch 30

(i) Epoch 40

(j) Epoch 50

Figure 13: Benchmark dataset two moons in 2D (see fig. 13a). Despite the presence of the outliers and the shape not being convex, the method learns a linear separation of these two moons (see fig. 13b). The loss values (loss $= \sum_{i=1}^{K} \lambda_i$) decrease consistently with the number of iterations(see fig. 13c). Optimizing kernel bandwidth $\sigma$ using gradient descent stabilizes as the $\sigma$ gradient diminishes (see fig. 13d). A snapshot of the optimized trajectory is in figs. 13e to 13j, where the first two eigenvectors (corresponding to the outliers) are omitted and the subsequent ones are visualized instead.

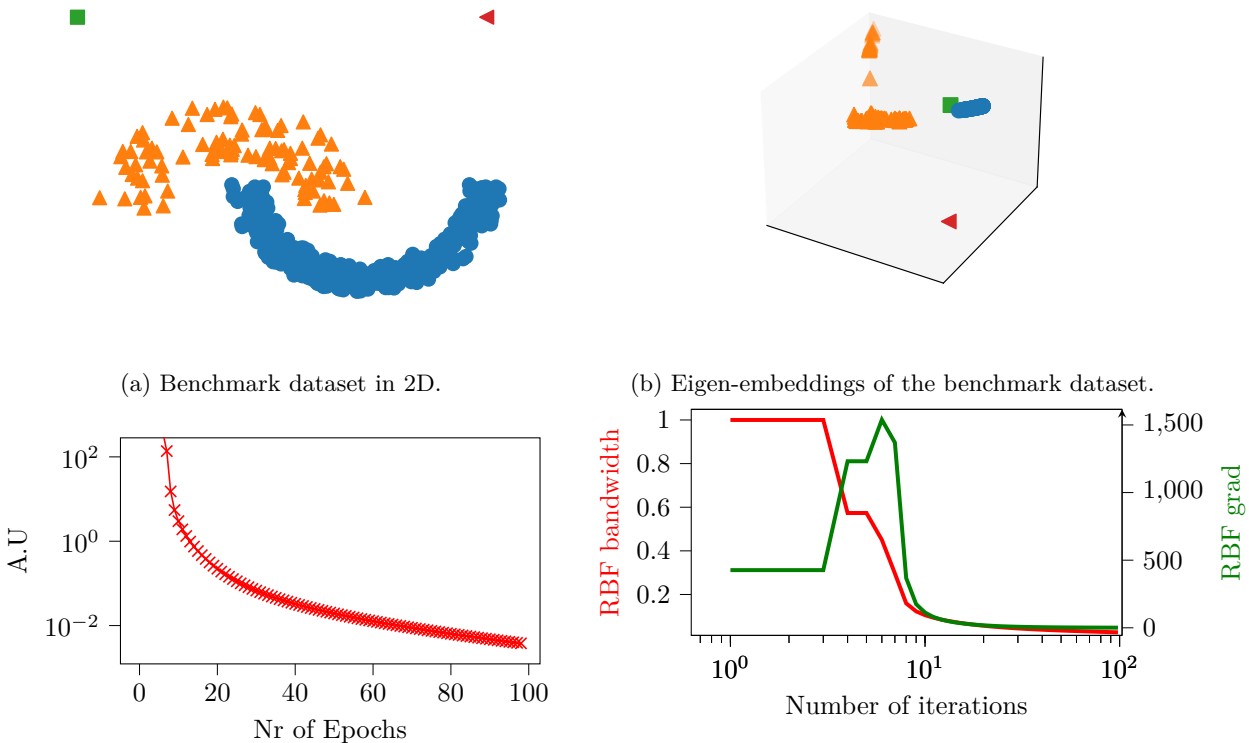

(a) Benchmark dataset in 2D.

(b) Eigen-embeddings of the benchmark dataset.

(c) Loss function (loss $= \sum_{i=1}^{K} \lambda_i$) over during training.

(d) RBF bandwidth $\sigma$ and its gradient during training.

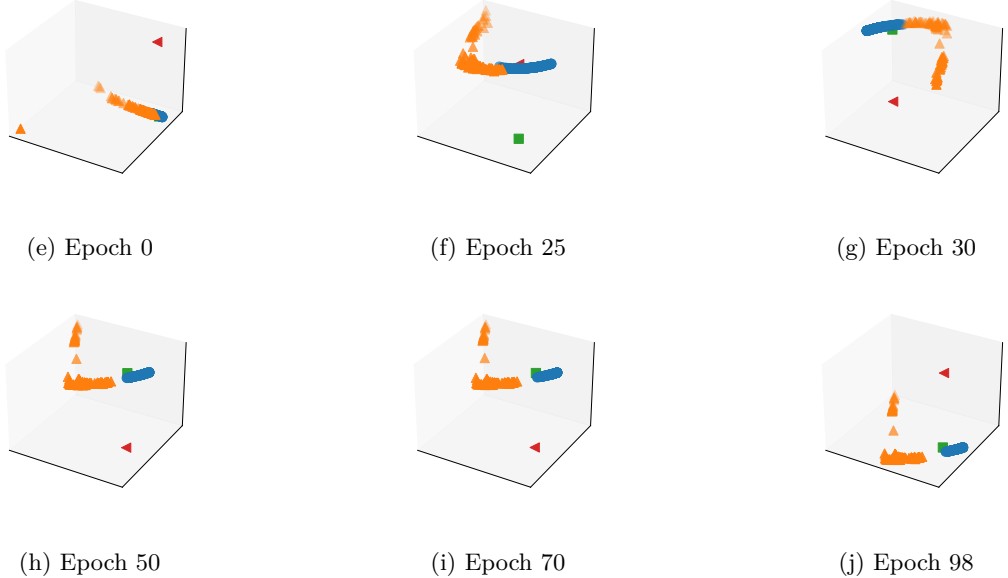

(e) Epoch 0

(f) Epoch 25

(g) Epoch 30

(h) Epoch 50

(i) Epoch 70

(j) Epoch 98

Figure 14: Benchmark dataset two (asymmetric) half moons in 2D (see fig. 14a). Despite the presence of the outliers and the shape not being convex, the method learns a linear separation of these two half moons (see fig. 14b). The loss values (loss $= \sum_{i=1}^{K} \lambda_i$) decrease consistently with the number of iterations (see fig. 14c). Optimizing kernel bandwidth $\sigma$ using gradient descent stabilizes as the $\sigma$ gradient diminishes (see fig. 14d). A snapshot of the optimized trajectory is in figs. 14e to 14j, where the first two eigenvectors (corresponding to the outliers) are omitted and the subsequent ones are visualized instead.

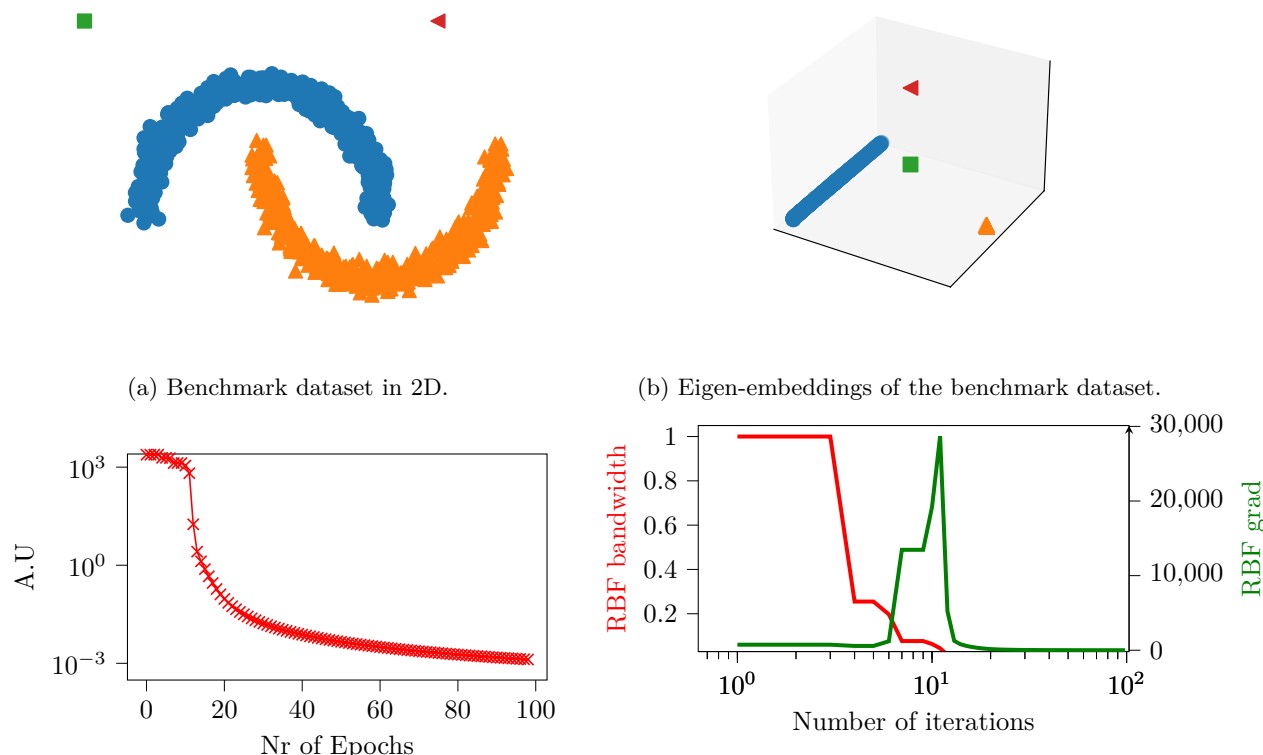

(a) Benchmark dataset in 2D.

(b) Eigen-embeddings of the benchmark dataset.

(c) Loss function (loss $= \sum_{i=1}^{K} \lambda_i$) over during training.

(d) RBF bandwidth $\sigma$ and its gradient during training.

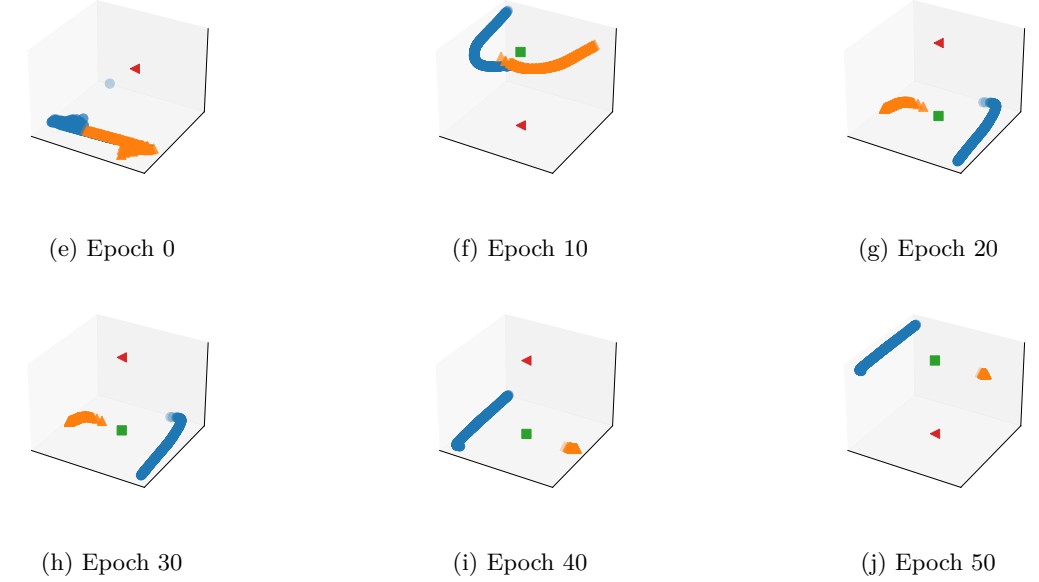

(e) Epoch 0

(f) Epoch 10

(g) Epoch 20

(h) Epoch 30

(i) Epoch 40

(j) Epoch 50

Figure 15: Benchmark dataset two moons in 2D (see fig. 15a). Despite the presence of the outliers and the shape not being convex, the method learns a linear separation of these two moons (see fig. 15b). The loss values (loss $= \sum_{i=1}^{K} \lambda_i$) decrease consistently with the number of iterations(see fig. 15c). Optimizing kernel bandwidth $\sigma$ using gradient descent stabilizes as the $\sigma$ gradient diminishes (see fig. 15d). A snapshot of the optimized trajectory is in figs. 15e to 15j, where the first two eigenvectors (corresponding to the outliers) are omitted and the subsequent ones are visualized instead.

## C  Optimization trajectory on the image and text datasets

For each tested dataset, we present the loss (loss $= \sum_{i=1}^{K} \lambda_i$) over iterations (see figs. 16a, 17a, 18a, 19a, 20a, 21a and 22a) together with the trajectory of the RBF bandwidth and its gradient (see figs. 16b, 17b, 18b, 19b, 20b, 21b and 22b). The convexity of the method persists for each dataset, as is evident from the always-decreasing magnitude of the RBF bandwidth gradient. Notice that although RBF bandwidth always starts at $\sigma = 0.5$, its starting gradient varies across different types of datasets.

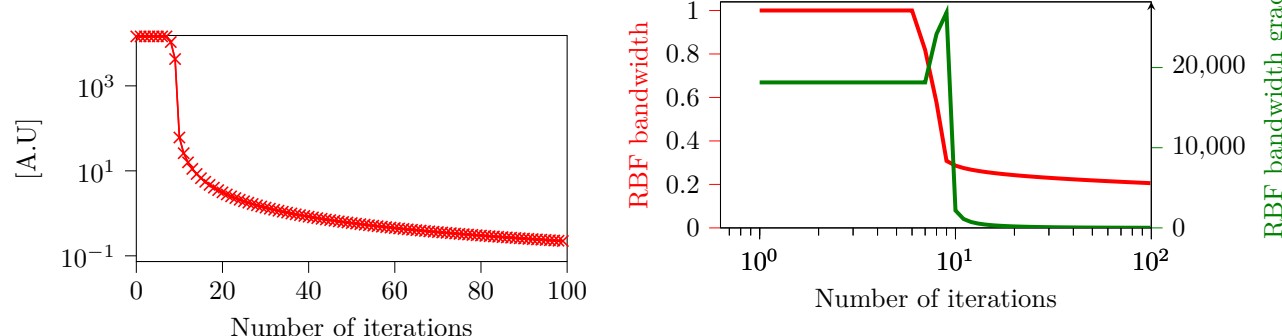

(a) Loss function (loss $= \sum_{i=1}^{K} \lambda_i$) over during training.   (b) RBF bandwidth $\sigma$ and its gradient during training.

Figure 16:   The loss values (loss $= \sum_{i=1}^{K} \lambda_i$) over the iterations for the COIL20 dataset in fig. 16a.Optimization of the $\sigma$ on the COIL20 dataset in fig. 16b.

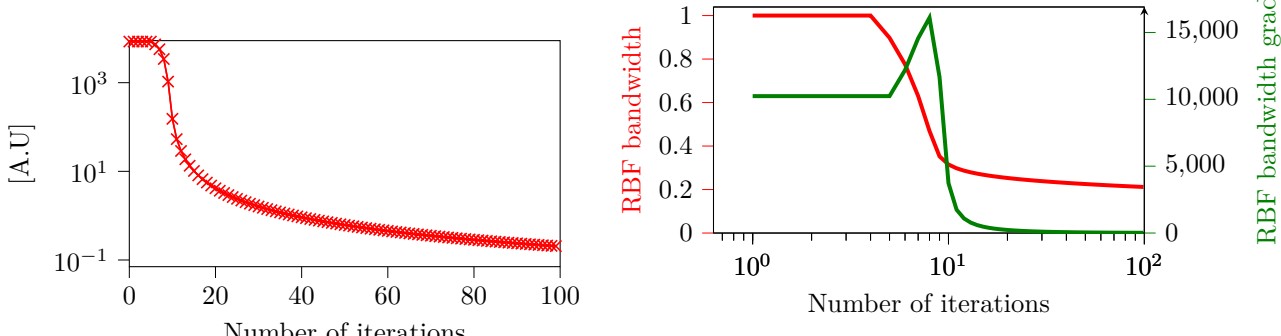

(a) Loss function (loss $= \sum_{i=1}^{K} \lambda_i$) over during training.   (b) RBF bandwidth $\sigma$ and its gradient during training.

Figure 17: The loss values (loss $= \sum_{i=1}^{K} \lambda_i$) over the iterations for the ORL dataset in fig. 17a.Optimization of the $\sigma$ on the ORL dataset in fig. 17b.

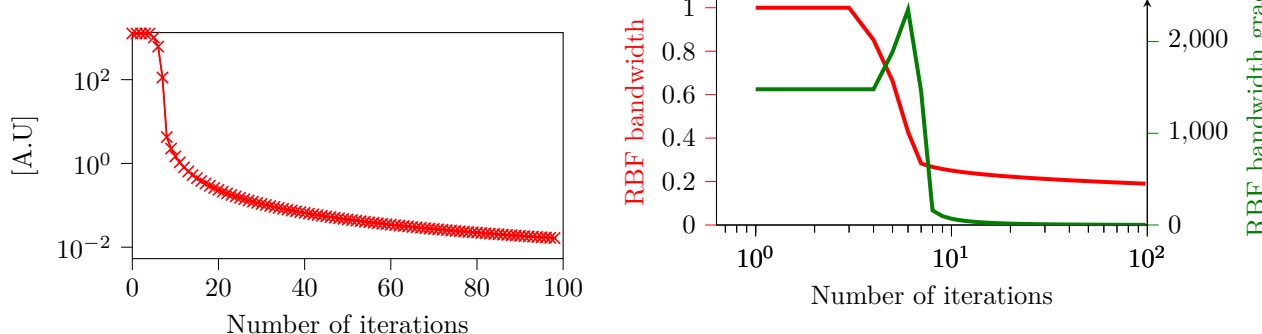

(a) Loss function (loss $= \sum_{i=1}^{K} \lambda_i$) over during training.   (b) RBF bandwidth $\sigma$ and its gradient during training.

Figure 18: The loss values (loss $= \sum_{i=1}^{K} \lambda_i$) over the iterations for the YALE dataset in fig. 18a. Optimization of the $\sigma$ on the YALE dataset in fig. 18b.

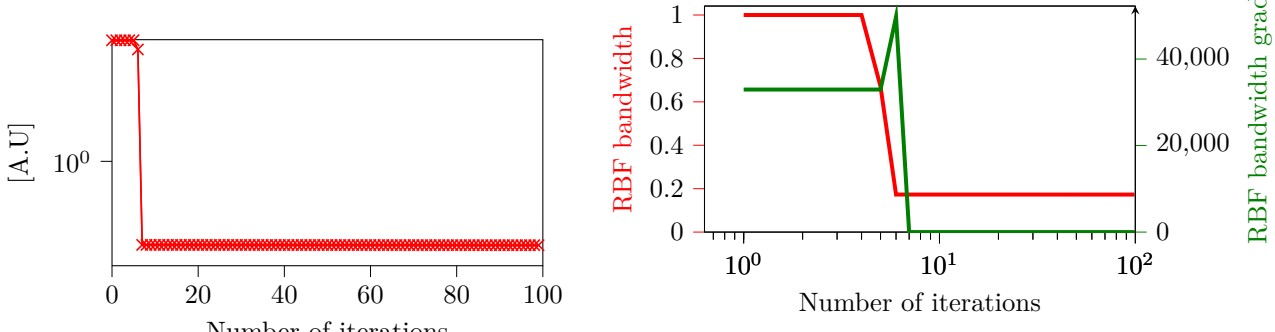

(a) Loss function (loss $= \sum_{i=1}^{K} \lambda_i$) over during training.   (b) RBF bandwidth $\sigma$ and its gradient during training.

Figure 19: The loss values (loss $= \sum_{i=1}^{K} \lambda_i$) over the iterations for the BA dataset in fig. 19a. Optimization of the $\sigma$ on the BA dataset in fig. 19b.

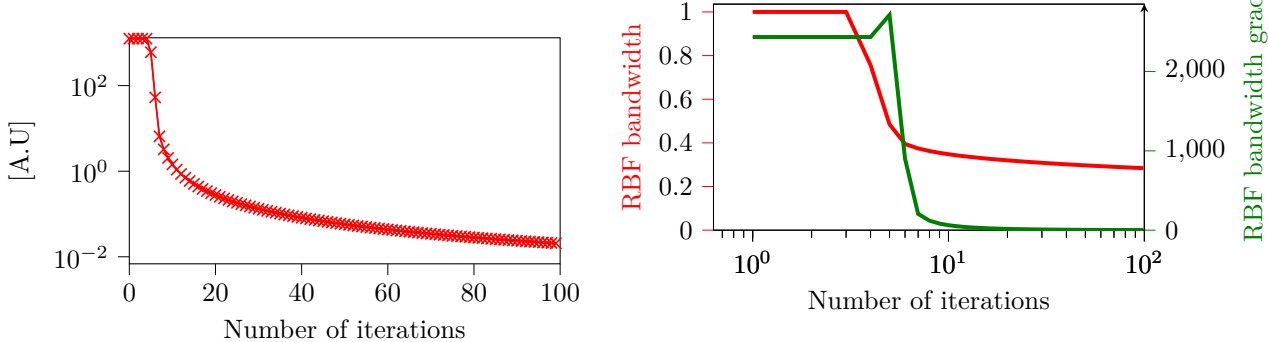

(a) Loss function (loss $= \sum_{i=1}^{K} \lambda_i$) over during training.   (b) RBF bandwidth $\sigma$ and its gradient during training.

Figure 20: The loss values (loss $= \sum_{i=1}^{K} \lambda_i$) over the iterations for the TR11 dataset in fig. 20a. Optimization of the $\sigma$ on the TR11 dataset in fig. 20b.

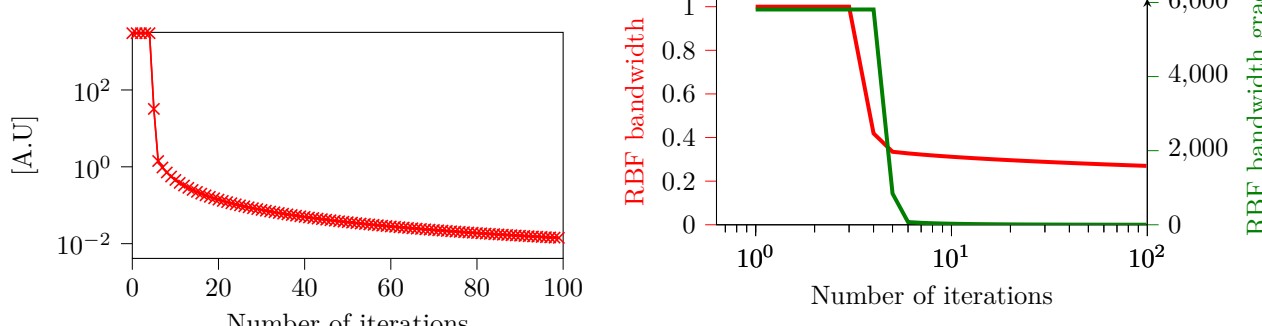

(a) Loss function (loss $= \sum_{i=1}^{K} \lambda_i$) over during training.   (b) RBF bandwidth $\sigma$ and its gradient during training.

Figure 21: The loss values (loss $= \sum_{i=1}^{K} \lambda_i$) over the iterations for the TR41 dataset in fig. 21a.Optimization of the $\sigma$ on the TR41 dataset in fig. 21b.

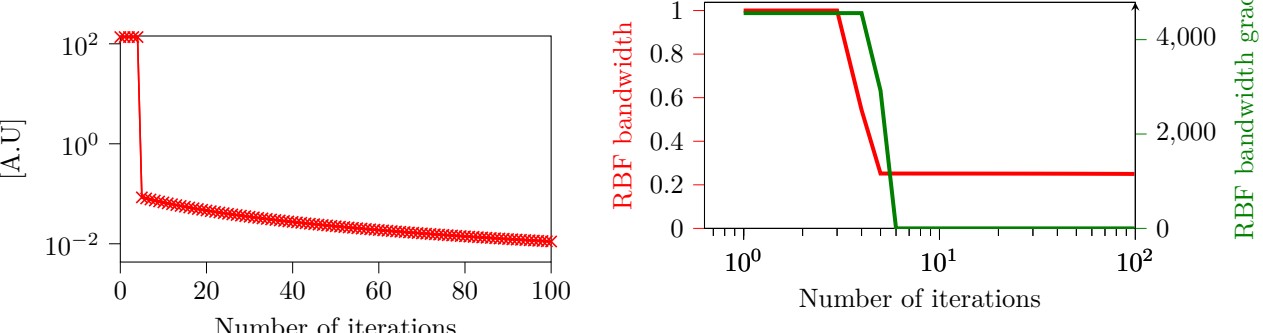

(a) Loss function (loss $= \sum_{i=1}^{K} \lambda_i$) over during training.   (b) RBF bandwidth $\sigma$ and its gradient during training.

Figure 22: The loss values (loss $= \sum_{i=1}^{K} \lambda_i$) over the iterations for the TR45 dataset in fig. 22a.Optimization of the $\sigma$ on the TR45 dataset in fig. 22b.

