# OpenReview forum: "Unsupervised Similarity Learning for Spectral Clustering"
_TMLR — Rejected by TMLR_

### Review · Reviewer_VxEx · 2024-04-22

**Summary Of Contributions:**

This article proposed an adaptive approach for determining hyperparameters in the similarity metric used for spectral clustering. The authors theoretically prove the convexity of the objective of spectral clustering within a certain range of the parameter \sigma, enabling the use of gradient descent with backpropagation for parameter optimization in each iteration. Experimental results on synthetic and real-world datasets demonstrate the effectiveness of the proposed approach.

**Audience:**

Yes

**Broader Impact Concerns:**

The broader impact section is not included in this article. Based on my assessment, there are no concerns regarding the ethical implications of this work that need to be addressed.

**Claims And Evidence:**

No

**Requested Changes:**

Please refer to the above-mentioned weaknesses.
1. The revised version should address or highlight Weaknesses 1, 2, and 3.
2. Please refer to references [1-5] and other relevant literature to enhance the literature review.
3. The proposed approach should be compared with the most recent and closely related studies.
4. The errors and ambiguities in the article need to be corrected and clarified.
5. Please polish the language expression and revise the presentation and organization of this article.

**Strengths And Weaknesses:**

Strengths:
1. The bandwidth parameter \sigma of the radial basis function (RBF) is treated as an optimization variable, aiming at adaptively finding its solution during optimization. Typically, \sigma needs to be tuned in RBF-based similarity graph learning for spectral clustering.
2. The convexity of the objective of spectral clustering is demonstrated within the interval (0, 1/2] through Theorem1, Lemmas 1, 2, 3, and Corollary 1, making it possible to optimize the parameter \sigma using gradient descent with backpropagation.
3. The proposed approach is evaluated through visualization experiments on synthetic datasets and quantitative experiments on real-world datasets.

Weaknesses:
1. One major concern arises from the limited scope of optimization within the narrow interval (0, 1/2]. The broader parameter space beyond (0, 1/2] remains unexplored. Albeit the objective of spectral clustering is non-convex within the extended range, there is potential to devise approximation algorithms to address this challenge. While these approximation algorithms may not yield the global optimal solution, they can still offer solutions that are close to optimal. This approach allows for the exploration of potential solutions across a wider parameter spectrum, and in certain scenarios, the approximate solutions may even surpass the optimal solutions found within the narrower range. Hence, the development of approximation algorithms for solving exact problems presents an intriguing and significant avenue for future research. Additionally, it’s worth noting that the term "optimal \sigma" used in the article is inaccurate, as the \sigma obtained through the proposed algorithm represents a locally optimal solution or a saddle point rather than a globally optimal solution.
2. Another major concern arises from the limited practicality and scalability of the proposed approach. While the adaptively optimized parameter \sigma during the similarity learning stage is a step forward, introducing additional regularization parameters in the clustering stage could potentially introduce complexities. For instance, integrating post-processing techniques like k-means or spectral rotation into the optimization objective may complicate the clustering process. Even if the clustering and spectral objectives are not jointly optimized, post-processing remains necessary with the proposed approach. Recent research has been focusing on end-to-end spectral clustering methods, as evidenced by the reference [1]. Moreover, the optimization of parameter \sigma in the proposed approach is based on the traditional spectral clustering model, which can be computationally expensive for handling large-scale datasets. Notably, the experiments conducted in this study utilize small to medium-sized real-world datasets. Recent research has been focusing on fast discrete spectral clustering methods, as evidenced by the reference [2]. Moreover, incorporating regularization or constraints associated with the RBF-based graph into the objective, such as sparse or low-rank regularization, presents challenges. Similarly, constructing the RBF-based graph in a dynamically learning low-dimensional space is complex. Overall, the all above factors limit its practicality and scalability.
[1] Peng, Y., Huang, W., Kong, W., Nie, F. and Lu, B. L., 2023. JGSED: An End-to-End Spectral Clustering Model for Joint Graph Construction, Spectral Embedding and Discretization. IEEE Transactions on Emerging Topics in Computational Intelligence.
[2] Nie, F., Lu, J., Wu, D., Wang, R. and Li, X., 2023. A Novel Normalized-Cut Solver with Nearest Neighbor Hierarchical Initialization. IEEE Transactions on Pattern Analysis and Machine Intelligence.
3. The convergence of the proposed Algorithm 1 cannot be guaranteed due to the heuristic strategy used to set the learning rate 'lr'. However, in Figures 4, 5, 6, 8, 9, and 11-16, it appears that Algorithm 1 converges, and in Figures 3 and 7, the gradient curves exhibit non-convergent behaviour since they initially descend, then rise, and finally descend again. An explanation for this phenomenon should be provided. Are the learning rates associated with the provided gradient curves fixed or not? Empirically, convergence is only guaranteed when 'lr' is fixed.
4. The literature review provided in Section 3 on Related Work appears to be incomplete. It mainly focuses on neighbourhood-based and self-expression-based approaches, while overlooking numerous other relevant studies. Related studies can be categorized into the following groups:
(1)	Empirical heuristic parameter assignment for RBF kernel, such as
[3] Zelnik-Manor, L. and Perona, P., 2004, December. Self-tuning spectral clustering. In Proceedings of the 17th International Conference on Neural Information Processing Systems (pp. 1601-1608).
(2)	Learning optimal value for parameter \sigma from data, such as
[4] Karasuyama, M. and Mamitsuka, H., 2013, December. Manifold-based similarity adaptation for label propagation. In Proceedings of the 26th International Conference on Neural Information Processing Systems-Volume 1 (pp. 1547-1555).
(3)	Locality-inducing graph learning;
(4)	Graph learning based on global reconstruction;
(5)	Sparsity-inducing graph learning;
(6)	Low-rank graph learning;
(7)	Graph learning in an adaptively transformed space.
For more details, please refer to the following paper:
[5] Qiao, L., Zhang, L., Chen, S. and Shen, D., 2018. Data-driven graph construction and graph learning: A review. Neurocomputing, 312, pp.336-351.
5. The latest comparison methods were published in 2019. The proposed approach should be compared with the most recent studies published within the last three years and closely related studies, as referenced in [1]-[5], or authors can conduct their own search for relevant literature.
6. There are a number of errors and ambiguities in this article.
(1)	In the RBF function, there is a “2” in front of the square of \sigma. The reason for removing the “2” in Eq. (1) should be explained in this article. It should be pointed out that Eq. (1) can be considered as a variant of the original RBF function. Furthermore, it should be noted in the context why it is necessary to express the Taylor series expansion of the RBF function, where the Taylor series expansion is the last term in Eq. (1).
(2)	The expression conveyed by Eq. (5) is confusing. It seems to suggest that because f^Tf=1, f^T\lambda(\sigma)f=\lambda(\sigma), right? However, this is incorrect. f^TL(\sigma)f should directly equal \lambda(\sigma), as dictated by the definition of eigenvalue decomposition.
(3)	In Theorem 1, the lowercase "k" should be changed to uppercase "K". In Eqs. (8)-(10), why is the variable "i" traversed starting from 0 instead of 1?
(4)	In Lemma 1, “incident matrix” should be “incidence matrix”. For a positive semi-definite matrix, there exists a real matrix such that the positive semi-definite matrix equals the product of the real matrix and its transpose. I believe here “incidence matrix” should be modified to “real matrix”.
(5)	The input and output of Algorithm 1 should be provided.
(6)	Eqs. (13) and (14) are incorrect. In Eq. (13), The summation symbol for variables i and j from 1 to N should precede both the first and second terms, while the summation symbol for variable k from 0 to K should not appear before the second term. In Eq. (14), there is a missing closing parenthesis, and the summation symbol for variable k from 0 to K should not appear before the second term. In Eqs. (13) and (14), constraints regarding Z can be added.
7. The overall language expression in this article requires significant revision. The current version is somewhat difficult to read.
8. The presentation and organization of this article could be further improved. For example, rearranging the order of Theorem 1 and Lemmas 1-3 would make them read more smoothly; Tables 1-4 would look better as a three-line table. In the current version, tables 1-4 are missing bottom borders, which looks somewhat odd; Changing the title of Section 5 from "Discussion" to "Conclusion" would be more appropriate.

---

> ### Author Response · Authors · 2024-05-17
> **Comments on the requested changes nr 4**
>
> Dear Reviewer,
>
> We are grateful for your detailed review and insights.
>
> The manuscript has been revised to reflect your suggestions with changes highlighted in the manuscript. Specifically:
>
> 1.We fixed the algorithm
>
> 2.Fixed the typos across the equations.

---

> ### Author Response · Authors · 2024-05-17
> **Comments on the requested changes nr  2 and 3**
>
> Dear Reviewer,
>
> We are grateful for your valuable recommendation to expand our survey to include a broader range of strategies. Following your constructive input:
>
> - We have enhanced the related work section by incorporating the approaches you suggested, as well as "Self-tuning spectral clustering." Advances in neural information processing systems 17 (2004).
>
> - We have created Table 3 for comparison, showcasing a selection of recent methods alongside our own proposed method for a clearer position.

---

> ### Author Response · Authors · 2024-05-18
> **Comments on the weaknesses 1 and 3**
>
> Dear Reviewer,
>
> We thank you very much for your time.
>
> We have updated our initial assertion concerning the convexity of the function in the interval (0, 1/2], following the insights shared by Reviewer Bxdf.
>
> This claim could not be substantiated for all cases within that range.
>
> To rectify this, we have redone our experiments, adjusting the optimization of $\sigma$ to consider the wider interval (0, 1] with an initial value of $\sigma=1$.

---

> ### Author Response · Authors · 2024-05-18
> **Comments on the requested changes nr 4**
>
> Dear Reviewer,
>
> We thank you for your efforts in thoroughly reviewing our paper.
>
> In response, we have revised portions of the text for clarity (please refer to the highlighted sections).

---

> ### Author Response · Authors · 2024-05-18
> **Comments on the joint training for Graph Learning, Discretization, and K-means**
>
> Dear Reviewer,
>
>
> Thank you for your valuable comments on our manuscript.
>
>
> Our approach builds upon the spectral clustering framework, necessitating post-processing steps. However, it's important to note that the loss function we've proposed acts as an effective surrogate for the simultaneous optimization of three key stages:
> 1. K-means clustering.
> 2. discretization, and
> 3. graph learning,
>
>
> This is because:
>
>
> - The optimization of the graph cut value $GC[A,f]=\sum_{i,j,j<i}A_{i,j}(f_{i}-f_{j})^{2}$ serves as an implicit stand-in for the K-means loss $K-means = \sum_{i=1}^{N}\sum_{k=1}^{K} r_{ik}||x_i - \mu_k||^2$. Essentially, minimizing our graph cut loss is akin to minimizing the K-means objective.
>
>
> - The optimization naturally yields the Laplacian matrix, whose eigenvectors correspond to minimal intra-cluster variations, thus fulfilling the role of discretization in partitioning the data.
>
>
> - Lastly, the learning process directly informs us about graph connectivity, thereby explicitly influencing the graph learning component.

---

> ### Author Response · Authors · 2024-05-18
> **Comments on the scalability of the proposed method.**
>
> Dear Reviewer,
>
> We appreciate the valuable comments you've provided regarding our research.
>
> We acknowledge scalability as an intrinsic challenge of spectral clustering (SC). In future work, we aim to tackle this issue by developing a sparse Laplacian matrix derived from a dense affinity matrix. Currently, our primary focus is on refining the bandwidth of the RBF kernel.

---

### Review · Reviewer_Bxdf · 2024-04-30

**Summary Of Contributions:**

The paper proposes to learn the bandwidth parameter for a Gaussian RBF similarity kernel for use within spectral clustering. The authors claim their approach, unlike existing approaches, is free of hyperparameters. Some experiments are provided which show potential in the proposed approach in some practical settings.

**Audience:**

Yes

**Broader Impact Concerns:**

Some of the derivations are well known. It is not clear if the authors are claiming that all of their derivations are novel, but if they are then this might constitute plagiarism.

**Claims And Evidence:**

No

**Requested Changes:**

I think that in order to continue with this paper, the motivation needs to be changed. The incorrect theory is not a death-knell since optimisation is over a single parameter and so non-convexity is not too much of an issue. Furthermore locally optimal solutions are justifiable. However, it is the case that the eigenvalues will be minimised as \sigma approaches zero and the solution will not be a sensible clustering solution, hence some sort of regularisation would need to be introduced, which sadly inevitably defeats the purpose of having a hyper-parameter free method.

**Strengths And Weaknesses:**

The main strength of the paper is that it tackles and very interesting and difficult problem, and the statements about the difficulty of validating selection of hyperparameters in an unsupervised context is extremely challenging

Unfortunately there is an overriding weakness in the paper which is that the theoretical results are not correct. Indeed it should be immediately clear that the convexity of the elements in the Laplacian with respect to the bandwidth must depend on the distances between data points, and so any result which suggests otherwise must be flawed. Indeed this can be seen by the fact that the proof of Lemma 3 is incorrect and the term in brackets in (12) should be (2 d(x, y)^2/\sigma^2 - 3), and so clearly this is positive/negative depending on the relative values of d(x, y) and \sigma.

Moreover, the primary objective is not well motivated. The Normalised Cut objective is sensible for clustering conditional on the similarities being representative of the data topology, but this does not mean that choosing the bandwidth such that the resulting Normalised Cut is small is well founded. In fact it can be shown that as the bandwidth tends to zero so too does the normalised cut and the solution is to separate clusters based on the maximum margin principle (maximum distance between clusters) [1]. This is not generally recommended in practice as it has no robustness at all to noise.

In addition there are numerous typographical and grammatical errors and mathematical objects not in the correct font, as well as numerous non-sequiturs; not to mention mathematical objects being used without being first defined. Finally, some of the derivations are very well known, e.g., the fact that the relaxed normalised cut is solved by the eigenvectors of the graph Laplacian.

[1] Hofmeyr, David P. "Connecting spectral clustering to maximum margins and level sets." Journal of Machine Learning Research 21.18 (2020): 1-35.

---

> ### Author Response · Authors · 2024-05-17
> **Comment regarding the convexity**
>
> Dear Reviewer,
>
> Thank you for pointing this out in our paper.
>
> We reviewed the theoretical results and reduced the claim regarding the convexity of the proposed loss w.r.t the RBF bandwidth.

---

> ### Author Response · Authors · 2024-05-17
> **Comment regarding sensible clustering as \sigma  goes towards zero**
>
> Dear Reviewer,
>
> Thank you very much for your insightful feedback.
>
> Indeed the presence of noise (such as outliers) can be detrimental to the performance of SC.
>
> Since these data are very distanced from the data their graph cut cost should be minimal.
>
> Since eigenvectors provide bi-partitioning of the dataset whose graph cut cost is given by the corresponding eigenvalue, these outliers will be accommodated on the first eigenvectors.
>
> While the core dataset is partitioned using the follow-up eigenvectors as their partitioning graph cut cost is higher than the outliers.
>
> Eventually, one can identify these outliers by checking the first eigenvectors sequentially and increasing the number of eigenvectors accordingly to enable the partition of the core dataset as well.
>
> For our method to function optimally, it is essential to identify these outliers and treat them as distinct clusters.
>
> This step is crucial to ensure that the spectral clustering method is accurately capturing the meaningful structure of the main data without undue influence from outliers.
>
> We experimented using the same diagnostic dataset by adding two outliers and included the results in Figures 4, and 11-15 in the Appendix.

---

> ### Author Response · Authors · 2024-05-17
> **Comment regarding typographical and grammatical errors**
>
> Dear Reviewer,
>
> We greatly appreciate your thoughtful feedback.
>
> In response, we have revised portions of the text for clarity (please refer to the highlighted sections), and we have corrected all the grammatical and typographical errors identified in the manuscript.
>
> Additionally, we have updated the formatting of our equations to comply with the guidelines provided by TMLR.
>
> We included the derivation for the normalized cut solution via EVD of the graph Laplacian to make the paper more self-contained and we referenced this part more thoroughly to indicate that is not part of our contribution.

---

> ### Comment · Reviewer_Bxdf · 2024-05-29
> **Reply to authors**
>
> Thanks to the authors for addressing my comments and concerns.
>
> I am afraid I still have some important reservations regarding the motivation behind the proposed objective. As mentioned in my initial review, the normalised cut (and so, indirectly, the eigenvalue(s) of the Laplacian) is a well motivated objective for clustering for a GIVEN affinity matrix, however that does not automatically mean that modifying the affinity matrix to reduce the normalised cut/eigenvalue(s) of Laplacian is well motivated. Indeed, as you have observed in your experiments, and which is theoretically verifiable, the objective you have adopted is decreasing towards zero as $\sigma$ goes to zero. As a consequence, if we accept the objective you proposed, we would conclude that for every data set it is appropriate to select as small a value of $\sigma$ "as possible" (noting of course, that there is no such smallest positive number).
>
> Moreover, if we were to adopt the objective you propose, I do not see the use of the optimisation approach discussed. We know the direction of the gradient will point us towards smaller and smaller values for $\sigma$, so we do not need to actually compute the gradient.
>
> I am afraid that I still do not find the objective to be well motivated, and more unfortunately I feel as though it is a self defeating endeavour to find it motivated since by accepting it as valid much of the remainder of the proposed method becomes redundant due to the knowledge that the objective decreases to zero as $\sigma \to 0$.

---

> > ### Author Response · Authors · 2024-05-30
> > **Commenting the motivation of the proposed objective.**
> >
> > Dear Reviewer,
> >
> > many thanks for your time and for bringing up this very important point.
> >
> > Setting the RBF bandwidth $\sigma$ infinitesimally small does result in a Graph Cut (GC) that is also infinitesimally small, and by extension also the gradient of $\sigma$.
> > However, the partition from the constructed Laplacian using this $\sigma$ does not reflect the topology of the data.
> > This is because exceedingly small values of $\sigma$ amplify the distances to such an extent that, after exponentiation, all the similarities become equally infinitesimally small.
> > Consequently, the associated affinity matrix values are uniform and infinitesimally small values rendering each data point essentially forming a singleton cluster.
> > As a result of this, the GC is minimal since all the eigenvalues are zero and not just the first K ones.
> >
> > Indeed by back-propagating with the proposed loss the obtained gradient of the RBF bandwidth $\sigma$ is always non-negative when $\sigma\in R_{\geq 0}$ however this does not mean that approaches zero infinitesimally using our poposed approach.
> > The gradient, however, approaches zero infinitesimally as indicated in the trajectories of $\sigma$ as the RBF bandwidth optimal value.
> > Thus, our approach provides a distinct RBF bandwidth $\sigma$ value that minimizes GC which is tailored to the topology, rather than defaulting to an arbitrarily small value irrespective of the data's inherent structure.
> >
> >
> >
> > We have provided a small example in the supplementary material to showcase the differences in eigen-embeddings between setting $\sigma$ manually very small and using the proposed approach.

---

> > > ### Comment · Reviewer_Bxdf · 2024-05-30
> > > **Reply**
> > >
> > > I agree that setting the bandwidth arbitrarily small is not a wise choice in practice, and I raised this only because of its absurdity. I just looked over your algorithm again and I do not see how this in any way learns a value appropriate for the topology of the data. Indeed, since we know the gradient is always positive your learning algorithm will decrease $\sigma$ either until iterations "run out" or it hits a negative value, in which case it will be increased again until it reaches a positive value, whereafter it will decrease again until iterations run out or it again goes negative. The final value arrived at appears to be dictated more by the number of iterations allowed than anything else. If I am wrong, please can you explain?

---

> ### Author Response · Authors · 2024-05-31
> **Comment regarding appropriate $\sigma$ value for the topology of the data**
>
> Dear Reviewer,
>
> many thanks for another very important aspect.
>
> Since the exponentiating the distances is an always non-decreasing trajectory one can say that for every pair of distance $d_{i}\geq d_{j}$  would be the same inequality even after their exponentiation. Thus RBF kernel would preserve the topology since the order of the pairwise distances would remain intact given that $\sigma$ does not become infinitesimally small.
>
> The reason why $\sigma$ stabilizes at a value that reflects the topology (and not just an infinitesimally small) while minimizing the GC is because the gradient of $\sigma$ researches zero much earlier than $\sigma$.  This is because as $\sigma$ goes down (almost) linearly (using gradient descent), the gradient itself decreases (almost) exponentially down to zero.
>
> The latter is linked to the fact that both GC and $\sigma$ go down. Since both $L(\sigma)$ and $\frac{\partial L(\sigma)}{\partial \sigma}$ are both symmetric Laplacian matrix and that $f_{1:K}$ are their first eigenvectors  $\lambda_{i}=f_{i}L(\sigma)f_{i}^{T}$,  $d\lambda_{i}=f_{i}dL(\sigma)f_{i}^{T}$, one can conclude that as we decrease GC on  $L(\sigma)$ simultaneously we minimize the GC in the $\frac{\partial L(\sigma)}{\partial \sigma}$ (aka minimize the gradient of $\sigma$).
>
> Simultaneosly, as we minimize $\sigma$ then the numerator overtakes the denominator since $\frac{\partial K(\sigma)}{\partial \sigma}=\frac{2d(x,y)e^{\frac{-d(x, y)}{\sigma^{2}}}}{\sigma^3}$.  Thus the similarity entries of the $\frac{\partial L(\sigma)}{\partial \sigma}$ decrease (almost) exponentially giving another minimization push for the gradient of $\sigma$.
>
> An $\sigma$ value appropriate for the topology of the data would mean that any existing data in the data would be linearized in the eigenspace. To do so, $\sigma$ decreases so that individual distances after exponentiation will be amplified leading to the bigger distances being even larger relative to the smaller ones. After normalization with the new amplified distances, the bigger distances would become even larger relative to the smaller distances and vice versa. This leads to the distances inside the existing gap would become uniformly large and the connection within each cluster would become uniformly small, aka linearization of the existing gap.  As this linearization in the data topology is attained, the gradient of $\sigma$ is minimized which prevents $\sigma$ from becoming smaller than necessary.

---

> ### Author Response · Authors · 2024-05-31
> **Comment regarding overshooting towards negative $\sigma$**
>
> Indeed, the RBF bandwidth $\sigma$ can hit negative values but this happens only at the beginning of the optimization.
> If this happens (line 16 in the Algorithm), then we reverse the update on $\sigma$ and decimate the learning rate (lines 17 and 18 in the Algorithm).
> The reason why this can happen only at the beginning is because the GC  $(i.e., \sum_{i=1}^{K}f_{i}L(\sigma)f_{i}^{T} )$ is very large and by extension also the gradient of $\sigma$ $(i.e., \sum_{i=1}^{K}f_{i}\frac{\partial L(\sigma)}{\partial \sigma}f_{i}^{T})$.
>
> Initially, once the learning rate stops being decimated, $\sigma$ starts decreasing along with GC.
> At the initial stages of the $\sigma$ update, it can be observed that the gradient of $\sigma$ might increase even though the GC on $L(\sigma)$ and $\sigma$ continue to decrease.
> This increase is possible as the similarity entries in $\frac{\partial L(\sigma)}{\partial \sigma}$ increase since the denominator is much smaller than its numerator for $\frac{\partial K(\sigma)}{\partial \sigma}=\frac{2d(x,y)e^{\frac{-d(x, y)}{\sigma^{2}}}}{\sigma^3}$ as $\sigma$ descended from its starting value of $\sigma<<=1$.
>
> This is a region where further decimations of the learning rate are possible depending on the dataset but unlikely as the learning rate has already been decreased at a geometric pace before any update on $\sigma$ even happens.
>
> Once the the gradient of $\sigma$ starts to decrease (almost) exponentially and the decimated learning rate decreases $\sigma$ (almost) linearly the risk of negative values is ruled out.
> Hence, you can safely say that $\sigma$ always stays inside the interval $(0,1]$.
>
> Yes, the number of iterations is important and has to be sufficiently high to accommodate the possible decimation of the learning rate and a sufficient amount of updates to ensure that the gradient approaches minimal values even though we get an exponential decrease in the gradient at an early stage.

---

> > ### Comment · Reviewer_Bxdf · 2024-05-31
> > **Further discussion**
> >
> > Thanks to the authors for elaborating on the trajectory of their algorithm. I don't disagree with what you have said, and that does at least better justify the algorithm as given. However, if I understand correctly Fig 3. c) is showing the trajectory of the loss function, and in this example how could we know not to have terminated  earlier, where in fact the gradient is not that small at the max number of iterations when compared with earlier in the learning. At the same time, there could equally be another such "plateau'' if we had chosen a larger number of iterations, or a different initial learning rate. As it is, it seems a lot is left to "chance" selection of the right number of iterations.
> >
> > Moreover, to return to my original comment on motivation, I still do not see how the objective of minimising the GC is fully justified. Since your algorithm actually doesn't seek to minimise the GC, but rather, if I understand your current argument, looks for the place where the gradient becomes extremely small (as this is a point at which the separation of clusters has been "learnt" by the setting of $\sigma$) the question of why you formulate the objective as minimising the GC is left begging.
> >
> > I think the argument from the point of view that when all the clusters have been "separated" by the appropriate setting of $\sigma$, further decreases do not reduce the GC substantially (i.e., the gradient becomes very small in magnitude), is workable. However, if that is the objective then I am very confident your algorithm could be sped up immensely just by evaluating the GC intelligently using some sort of bisection type approach following a rough grid search, rather than by a slow gradient descent scheme as presented.
> >
> > I'm afraid that currently there is inconsistency in the way the work is presented since you propose to minimise GC, but actually don't intend doing so.

---

> > > ### Author Response · Authors · 2024-06-08
> > > **The manuscript has been revised to include motivation for minimizing the Graph Cut (GC) by applying gradient descent to the parameter $\sigma$.**
> > >
> > > Dear Reviewer,
> > >
> > > following our discussion on the motivation of GC minimization leading to gradient descent on $\sigma$ we updated the manuscript with the new highlight around equation 7.
> > >
> > > Thank you for your time and please let us know if there are any further questions.

---

> > > > ### Comment · Reviewer_Bxdf · 2024-06-11
> > > > **Motivation still not evident**
> > > >
> > > > Thanks to the authors for further engagement. I have just looked again over the text in the latest version, but there still seems to be inconsistency. The objective is still written as minimising the GC as a function of \sigma, but as we have discussed in previous comments you actually do not seek to minimise the GC. Indeed, in the newly added text it still states "To minimize the GC loss...", and yet the GC loss is not minimised and nor is it the intention of the authors to actually minimise this quantity

---

> > > > > ### Author Response · Authors · 2024-06-11
> > > > > **Updated the motivation following the previous disucussion**
> > > > >
> > > > > Dear Reviewer,
> > > > >
> > > > > we thank you very much for your engagement as it is highly appreciated.
> > > > >
> > > > > We updated the text to highlight the derivation of the gradient descent on $\sigma$ in accordance with our discussion.
> > > > > The update in the manuscript (in the updated pdf) is now as follows:
> > > > >
> > > > > Whereby, we incrementally fine-tuned $\sigma$ using discrete steps of magnitude $\Delta\sigma$, which were determined through a linear approximation near $\sigma$ for $\sum_{i=1}^{K}\lambda_{i}(\sigma)$ (see first equation). Optimal adjustment of $\sigma$ is achieved when we choose $\Delta\sigma$ to be in the opposite direction to $\frac{\partial\sum_{i=1}^{K}\lambda_{i}(\sigma)}{\partial\sigma}$ (see second equation).
> > > > >
> > > > > \begin{equation}
> > > > >    \sum_{i=1}^{K}\lambda_{i}(\sigma+\Delta\sigma)\approx\sum_{i=1}^{K}\lambda_{i}(\sigma)+\Delta\sigma\frac{\partial\sum_{i=1}^{K}\lambda_{i}(\sigma)}{\partial\sigma}
> > > > > \end{equation}
> > > > >
> > > > > \begin{equation}
> > > > > \Delta\sigma=-\eta\frac{\partial\sum_{i=1}^{K}\lambda_{i}(\sigma)}{\partial\sigma}\to \sigma_{i}=\sigma_{i}-\eta\frac{\partial\sum_{i=1}^{K}\lambda_{i}(\sigma)}{\partial\sigma}
> > > > > \end{equation}

---

> > > > > > ### Comment · Reviewer_Bxdf · 2024-06-12
> > > > > > **Reply**
> > > > > >
> > > > > > Thanks again to the authors for their engagement.
> > > > > >
> > > > > > Yes, I had seen the text describing the gradient descent. My concern, however, is not about the implementation of the method, but the motivation for minimising the GC.

---

> > > > > > > ### Author Response · Authors · 2024-06-12
> > > > > > > **Changed the motivation**
> > > > > > >
> > > > > > > Dear Reviewer,
> > > > > > >
> > > > > > > many thanks for your time in reviewing and discussing the article.
> > > > > > >
> > > > > > > We altered the text which contained minimizing the GC and also added in the method section the motivation as:
> > > > > > >
> > > > > > > Therefore, the goal of the optimization process is to minimize $\sigma$ until its gradient approaches a negligibly small magnitude.
> > > > > > >
> > > > > > > Please, find attached the updated manuscript with other highlights in orange.

---

> ### Author Response · Authors · 2024-06-03
> **Further Discussion on the motivation.**
>
> Dear Reviewer,
>
> many thanks for your insightful feedback and your time.
>
> As you described it "all the clusters have been "separated" by the appropriate setting of $\sigma$, further decreases do not reduce the GC substantially (i.e., the gradient becomes very small in magnitude)", is precisely our objective.
>
> However, since minimizing $\sigma$ is guided by GC minimization therefore we wrote the former causes the latter.
> Despite the causation direction (i.e.,., minimizing $\sigma$-> minimizing GC) minimizing GC is an objective measure of the clustering (i.e., graph partitioning), and minimizing $\sigma$ alone cannot be objectively evaluated without GC.
>
> Moreover, we believe that it is important to emphasize (probably even add it to the manuscript) that there is a difference between minimizing GC and a minimal GC when it comes to producing sensible clusters.
> The latter is not what we are after as we can set $\sigma$ substantially small which would yield a small GC (and small gradient of $\sigma$).
> However, by doing this, there is a great risk of destroying the topology of the data and producing all eigenvalues zero.
> On the other hand, starting from a position of high GC (i.e., $\sigma=1$) means that all the edges are unchanged, thus the topology is intact.
> As we gradually reduce $\sigma$  (by minimizing GC as they go hand in hand) we gradually 'prune' the bigger distances sequentially until no $\sigma$ gradient (becomes very close to zero) does not change GC substantially while the 'topology remains intact'.
> The latter is what we aim for by minimizing $\sigma$ ( and hand-in-hand with the GC).
>
> The number of iterations is not detrimental to the performance as far as it is sufficiently high such as both GC and $\sigma$ gradient stabilizes at a small value. Moreover, unlike other unsupervised methods whose progress is solely tracked via the loss function, we can also utilize the gradient of the $\sigma$ to decide the termination of the training.
>
> Indeed, the speed of the performance can be improved as EVD is quite costly however, however, it provides us with the gradient, the direction of the fast change for the loss. Nevertheless, we are actively investigating for better speed and scalability.

---

### Review · Reviewer_fhn4 · 2024-05-12

**Summary Of Contributions:**

The submission presents a method for estimating the RBF width parameter used for similarity metric and spectral clustering, trying to "emphasise any existing separation within the data points" by lowering "the smaller pairwise distances" and amplifying "the more pronounced distances". The proposed method was shown to outperform several existing approaches on some benchmark datasets.

**Audience:**

Yes

**Claims And Evidence:**

No

**Requested Changes:**

In addition to the concerns listed above which prefers improvement, it seems to me that the work is still away from being mature enough to be published in a top-quality academic journal.

**Strengths And Weaknesses:**

Strengths:
1. Meaningful problem to study: Both metric learning and spectral clustering are important machine learning tasks. It is well-known that the quality of spectral clustering would suffer significantly from an inappropriately chosen width parameter if RBF kernel is used to construct the similarity metric. Despite the importance, so far there seems no well accepted methods to estimate the width parameter with theoretical justification, and instead empirical methods of manually choosing the parameter value is widely used. Therefore the study along this line is definitely meaningful even though related models (spectral clustering) has been studied more than two decades.

2. The developed algorithm in the submission seems not difficult to understand with a reasonable complexity.

Concerns:
1. Numerous writing inconsistencies/mistakes/typos, too much to enumerate fully. Just list a few in Algorithm 1 at Page 6.

    --Lines 4/13/14/15/16/17: The usage of $\sigma$, $\sigma_i$ and $\sigma_{i+1}$ seems a mess. Needs to be consistent in the algorithm description for the benefit of readers.

    --Line 8: Should be $\sigma^2$, not $\sigma$.

    --Line 13: Should be $\sum_{i=1}^{K}$, not $\sum_{i=0}^{K} $.

2. Important literature/comparison missing which makes the empirical results not convincing. Just list two:

    --The local scaling strategy suggested in Zelnik-Manor, Lihi, and Pietro Perona. "Self-tuning spectral clustering." Advances in neural information processing systems 17 (2004).

    --The automatic $\sigma$ selection strategy suggested in Ng, Andrew, Michael Jordan, and Yair Weiss. "On spectral clustering: Analysis and an algorithm." Advances in neural information processing systems 14 (2001).

    --The authors are suggested to do a more comprehensive survey and include representative strategies of setting $\sigma$ in their evaluation comparisons.

3. What will happen to if the optimal $\delta > \frac{1}{2}$, or $\frac{\partial L(\sigma)}{\partial \sigma}<0$ from the first iteration of $i$? It seems that Algorithm 1 will fail based on its description at Page 6. For theoretical completeness, this special case should be addressed.

---

> ### Author Response · Authors · 2024-05-17
> **Comment regarding concern 1**
>
> Dear Reviewer,
> We have updated the manuscript accordingly. In particular:
> - Inconsistencies with the usage of $\sigma$, $\sigma_i$, and $\sigma_{i+1}$ addressed throughout Lines 4, 13, 14, 15, 16, and 17 have been put to $\sigma_i$
> - On Line 8, we corrected $\sigma^2$ to correctly represent the squared scaling parameter .
> - Lastly, on Line 13, the summation index has been amended to begin at 1.
> We have highlighted the changes in the revised version of the manuscript.

---

> ### Author Response · Authors · 2024-05-17
> **Comment regarding concern 2**
>
> Dear Reviewer,
> Thank you for your insightful suggestion to conduct a more extensive survey that encompasses a wider array of strategies. In response to your feedback:
> - We have updated the related work section to feature the methodologies you highlighted, along with additional contributions recommended by Reviewer VxEx.
> - We have created Table 3 for comparison, showcasing a selection of recent methods alongside our own proposed method for a clearer position.

---

> ### Author Response · Authors · 2024-05-17
> **Comment regarding concern 3**
>
> Dear Reviewer,
>
> Thank you for highlighting this critical aspect of our algorithm. We acknowledge that the condition $\frac{\partial L(\sigma)}{\partial \sigma}<0$ is not applicable when $\sigma\in \mathbb{R}_{\geq 0}$, and we have addressed this by providing a proof in Theorem 1 within the Appendix.
>
> Additionally, we have updated our initial assertion concerning the convexity of the function in the interval (0, 1/2], following the insights shared by Reviewer Bxdf. This claim could not be substantiated for all cases within that range. To rectify this, we have redone our experiments, adjusting the optimization of $\sigma$ to consider the wider interval (0, 1] with an initial value of $\sigma=1$.

---

### Author Response · Authors · 2024-05-21
**Summary on the changes.**

Dear Reviewer

many thanks for all your insightful comments.
We made the required changes and amended the text to increase the clarity and readability.

To summarize the key modifications:

- We have expanded Table 3 to include comparisons with newer methodologies.
- Rewrote the related method section which includes the more recent works
- We have conducted additional tests concerning outlier data, detailing the results in Figures 4 and 11-15 in the Appendix to better illustrate the data partitioning.
- We have withdrawn the statement regarding convexity, in line with Reviewer Bxdf's recommendation.
- We have appended Theorem 1 in the Appendix, which provides a proof for the non-negativity of the RBF kernel's gradient within the non-negative real domain  $R_{\geq 0}$.

---

### Decision · Action_Editor_88wj · 2024-08-01

**Recommendation:** Reject

**Comment:**

Two of the three reviewers recommended to reject the paper.  The third reviewer never produced a recommendation despite repeated requests.  However, despite not receiving a response from the third reviewer, there are enough concerns raised by the other two reviewers in their reject recommendations to move forward with rejecting the paper.  Furthermore, the third reviewer noted in their original review that "it seems to me that the work is still away from being mature enough to be published in a top-quality academic journal."

The two recommendations were made after the revision was posted, and the reviewers noted several issues with that paper that still remained.  The following points were raised by the reviewers after the revision was posted:

1. The value range of \sigma has been updated to (0,1] from (0,1/2]. The assertion regarding the convexity of the proposed objective function in the interval (0,1/2] has been omitted. Given that the objective function cannot be ensured to be convex over either (0,1/2] or (0,1], what rationale underlies the utilization of gradient descent as the optimization algorithm?
2. In the following literature, it is noted that given mild assumptions on the similarity function, as the scaling parameter \sigma is reduced to zero, the spectral clustering solution converges to the maximum margin clustering solution. It would be beneficial to demonstrate that the Gaussian kernel employed in this study adheres to the mild assumptions delineated in the literature. [1] Hofmeyr D P. Connecting spectral clustering to maximum margins and level sets. Journal of Machine Learning Research, 2020, 21(18): 1-35.
3. In the experiments, outliers were intentionally introduced into the datasets. How can we ensure that the real-world datasets utilized in this study do not inherently contain outliers?
4. This paper utilizes the smallest eigenvectors to identify outliers, a common approach. However, eigenvectors corresponding to the smallest eigenvalues typically exhibit greater fluctuations and are more sensitive to noise. Relying solely on the smallest eigenvectors for outlier detection does not guarantee the identification of all outliers and may lead to misjudgments. Therefore, in practice, it is common to set a threshold to determine whether a node is an outlier, and selecting an appropriate threshold is a challenging task.
5. There are still some unresolved concerns from the previous version. Here are some issues, and there are others as well. For instance, the phrasing regarding the ‘optimal \sigma’ is inaccurate; Equation (5) is incorrect. Additionally, the convergence of the proposed algorithm cannot be guaranteed because of the heuristic approach employed to set the learning rate.
6. In the response, while spectral clustering and k-means indeed share some association, they have distinct optimization objectives. A decrease in the objective value of spectral clustering does not guarantee a decrease in the objective value of k-means. Even though the eigenvectors of the Laplacian matrix correspond to minimal intra-cluster variations, they still need to be discretized to obtain clustering results, right?
7. I think the motivation for the proposed approach is still lacking, and at best there is an important inconsistency in that the proposal is to minimise the eigenvalue objective which is known to approach zero as the tuning parameter approaches zero. As such, despite this being their proposed objective the authors do not actually intend to minimise it.

It seems that there are still several issues that need to be tackled before this paper is ready for publication.

**Audience:**

Yes, this paper is on a topic that is relevant to the TMLR audience.  Though spectral clustering is an area that attracted more attention several years ago than it does today, it is still a relevant topic area and continues to be studied.

**Claims And Evidence:**

All three reviewers have noted several issues with the claims and evidence of the paper.  Please see the comments section below, where I highlight some of the issues that the reviewers had after the revision was posted.  In particular, they noted issues with the motivation of the method, the experiments performed, and the correctness of some of the technical claims in the paper.  There are enough concerns here to suggest that the paper is not yet ready for publication.